# Uncertainty-aware Constraint Inference in Inverse Constrained Reinforcement Learning

**Sheng Xu**
School of Data Science
The Chinese University of Hong Kong, Shenzhen
`shengxu1@link.cuhk.edu.cn`

**Guiliang Liu**[*]
School of Data Science
The Chinese University of Hong Kong, Shenzhen
`liuguiliang@cuhk.edu.cn`

## Abstract

Aiming for safe control, Inverse Constrained Reinforcement Learning (ICRL) considers inferring the constraints respected by expert agents from their demonstrations and learning imitation policies that adhere to these constraints. While previous ICRL works often neglected underlying uncertainties during training, we contend that modeling these uncertainties is crucial for facilitating robust constraint inference. This insight leads to the development of an Uncertainty-aware Inverse Constrained Reinforcement Learning (UAICRL) algorithm. Specifically, 1) aleatoric uncertainty arises from the inherent stochasticity of environment dynamics, leading to constraint-violating behaviors in imitation policies. To address this, UAICRL constructs risk-sensitive constraints by incorporating distributional Bellman updates into the cumulative costs model. 2) Epistemic uncertainty, resulting from the model's limited knowledge of Out-of-Distribution (OoD) samples, affects the accuracy of step-wise cost predictions. To tackle this issue, UAICRL develops an information-theoretic quantification of the epistemic uncertainty and mitigates its impact through flow-based generative data augmentation. Empirical results demonstrate that UAICRL consistently outperforms other baselines in continuous and discrete environments with stochastic dynamics. The code is available at `https://github.com/Jasonxu1225/UAICRL`.

## 1 Introduction

Reinforcement Learning (RL) is an effective technique for solving sequential decision-making problems, typically focusing on maximizing cumulative rewards. However, recent studies (Liu et al., 2021; Satija et al., 2020; Yang et al., 2023) have argued that, to achieve safe control, the optimal policy must adhere to the underlying constraints in the environment. For instance, in an open-road environment, an autonomous driving policy must comply with traffic rules and social norms. Explicitly specifying these constraints is challenging. A more practical alternative is to infer the constraints from expert demonstrations by analyzing the behavioral patterns of expert agents.

To achieve this goal, Inverse Constrained Reinforcement Learning (ICRL) (Malik et al., 2021) extends the Maximum Entropy Inverse Reinforcement Learning (MEntIRL) framework (Ziebart et al., 2008) to infer constraints (rather than rewards) from expert demonstrations. ICRL alternates between Constrained Reinforcement Learning (CRL) and Inverse Constraint Inference (ICI) until the imitation policy can reproduce the expert demonstrations. During the process, traditional ICRL algorithms (Scobee & Sastry, 2020; Malik et al., 2021; Gaurav et al., 2023; Liu & Zhu, 2022; Qiao et al., 2023) often assume deterministic training environments, without considering the influence of underlying uncertainties. Specifically, in the CRL phase, stochastic transition functions introduce *aleatoric uncertainty*, thereby influencing the policy update in CRL and may result in constraint-violating behaviors. In pursuit of safety control, human experts often exhibit risk-averse behaviors, so an imitating agent cannot accurately replicate expert demonstrations unless it formulates a risk-sensitive policy. In the ICI phase, due to the finite size of the training data, *epistemic uncertainty* arises when game contexts lie outside the data distribution, leading to inaccurate cost predictions.

To achieve uncertainty-aware constraint inference, Liu et al. (2023); Papadimitriou et al. (2023) proposed modeling the posterior distribution of step-wise constraints using variational inference and

---

[*]Corresponding author: Guiliang Liu, liuguiliang@cuhk.edu.cn

Monte Carlo sampling. McPherson et al. (2021); Baert et al. (2023) incorporated maximum causal entropy likelihood into ICRL. Although causal entropy and constraint distribution are sensitive to aleatoric and epistemic uncertainties respectively, none of the previous methods can handle both uncertainties.

In this paper, we introduce the Uncertainty-aware Inverse Constrained Reinforcement Learning (UAICRL), a novel ICRL framework that models both the aleatoric and epistemic uncertainties for achieving robust constraint inference. Figure 1 provides an overview of UAICRL. Specifically,

1) We design a risk-sensitive constraint by modeling the distribution of cumulative costs and distorting this distribution to represent risk measures. The predicted distribution is sensitive to aleatoric uncertainty and we empirically justify these findings in our experiments. To enable efficient risk-sensitive control in continuous spaces, we propose the Distributional Lagrange Policy Optimization (DLPO), which incorporates distributional estimation and Lagrange mechanics into the classic Proximal Policy Optimization (PPO) (Schulman et al., 2017) for constrained policy optimization.

2) We introduce a mutual-information-driven metric (Gabrié et al., 2018) to quantify epistemic uncertainty in constraint inference and propose an information-theoretic ICI objective to minimize the impact of epistemic uncertainty when updating the constraint function. A key technique to achieve this objective involves augmenting the training data using the proposed Flow-based Trajectory Generation (FTG) algorithm. FTG effectively generates a diverse set of trajectories based on the dataset and task-dependent rewards, thus reducing the influence of OoD state-action pairs in constraint prediction.

Empirical evaluations show that UAICRL consistently achieves higher feasible rewards and lower constraint violation rates in stochastic environments with both discrete and continuous state spaces, outperforming other ICRL baselines. For a comprehensive evaluation, we conduct in-depth studies to individually assess how effectively UAICRL handles the aleatoric and epistemic uncertainty.

## 2 PROBLEM FORMULATION

**Constrained Reinforcement Learning (CRL).** To solve a CRL problem, the agent optimizes the control policy under a Constrained Markov Decision Processes (CMDPs) $\mathcal{M}^c$, which can be defined by a tuple $(\mathcal{S}, \mathcal{A}, p_{\mathcal{T}}, p_{\mathcal{R}}, p_{\mathcal{C}}, \epsilon, \mu_0, \gamma, T)$ where: 1) $\mathcal{S}$ and $\mathcal{A}$ denote the space of states and actions. 2) $p_{\mathcal{T}}(s'|s, a)$ and $p_{\mathcal{R}}(r|s, a)$ define the transition and reward probabilities. 3) $p_{\mathcal{C}}(c|s, a)$ and $\epsilon$ denote the probability of cost and the associated bound. 4) $\mu_0$ defines the initial state distribution. 5) $\gamma \in (0, 1)$ is the discount factor and $T$ defines the planning horizon ($T = \infty$ in principle). The goal of the CRL policy $\pi$ is to maximize expected discounted rewards under the constraint:

$$\arg\max_{\pi} \mathbb{E}_{\pi, p_{\mathcal{T}}, p_{\mathcal{R}}, \mu_0} \Big[ \sum_{t=0}^{T} \gamma^t r(s_t, a_t) \Big] \text{ s.t. } \mathbb{E}_{\pi, p_{\mathcal{T}}, p_{\mathcal{C}}, \mu_0} \Big[ \sum_{t=0}^{T} \gamma^t c(s_t, a_t) \Big] \leq \epsilon \tag{1}$$

Note that traditional CRL problems often assume the constraint signals are directly observable, but in real-world problems, such constraint signals are not readily available, and we must infer these constraints from the environment by solving the following inverse problem.

**Inverse Constraint Inference (ICI).** ICRL algorithms typically assume that *reward signals are observable and the goal is to infer the constraints*. Inspired by Malik et al. (2021), we define $\Phi$ as a Bernoulli feasibility variable that takes two values $\{\phi^+, \phi^-\}$ such that $p(\phi^+|s, a; \omega)$ ($\omega$ denotes model parameters) quantifies to what extent performing action $a$ in the state $s$ is feasible while $p(\phi^-|s, a; \omega)$ denotes the probability this movement is infeasible. For clarity, in the rest of this paper, we denote $\phi^+$ by $\phi$. Accordingly, the step-wise cost is then defined as $c_{\omega}(s, a) = 1 - p(\phi|s, a; \omega)$. Under these definitions, the likelihood of generating the expert dataset $\mathcal{D}_e$ can be represented as:

$$p(\mathcal{D}_e|\Phi) = \frac{1}{(\mathbb{Z}_{\mathcal{M}^{c_\omega}})^N} \prod_{n=1}^{N} \exp \Big[ r(\tau^{(n)}) \Big] \mathbb{1}^{\mathcal{M}^{c_\omega}}(\tau^{(n)}) \tag{2}$$

where 1) $\mathcal{M}^{c_\omega}$ denotes the CMDP with constraint function $c_\omega$, 2) $N$ denotes the number of trajectories in the expert dataset, 3) the normalizing term $\mathbb{Z}_{\mathcal{M}^{c_\omega}} = \int \exp[r(\tau)] \mathbb{1}^{\mathcal{M}^{c_\omega}}(\tau) d\tau$, and 4) the identifier $\mathbb{1}^{\mathcal{M}^{c_\omega}}(\tau^{(n)})$ can be defined as $p(\phi|\tau^{(n)}; \omega) = \prod_{t=1}^{T} p(\phi|s_t^{(n)}, a_t^{(n)}; \omega)$. By substituting them to Equation (2), we can update $\omega$ by the gradient of the log-likelihood function (Malik et al., 2021):

$$\nabla_{\omega} \log[p(\mathcal{D}_e|\Phi)] = \sum_{n=1}^{N} \Big[ \nabla_{\omega} \sum_{t=0}^{T} \log[p(\phi|s_t^{(n)}, a_t^{(n)}; \omega)] \Big] - N\mathbb{E}_{\hat{\tau} \sim \pi_{\mathcal{M}^{c_\omega}}} \Big[ \nabla_{\omega} \sum_{t=0}^{T} \log[p(\phi|\hat{s}_t, \hat{a}_t; \omega)] \Big] \tag{3}$$

where $\hat{\tau}$ is sampled from the current imitation policy $\pi_{\mathcal{M}^{c_\omega}}$. Intuitively, our goal is to differentiate the trajectories generated by expert policies and imitation policies that may violate the constraints. When the imitation policy under the learned constraint model $c_\omega$ matches the expert policy, this gradient goes to 0, and the update stops.

**Modeling Uncertainty in ICRL.** An ICRL agent iteratively solves the CRL and ICI problems until the imitation policy can reproduce expert demonstrations. However, existing ICRL algorithms typically assume a deterministic setting without considering uncertainties, but we argue that the aleatoric and epistemic uncertainties can emerge and significantly affect the training and the convergence of ICRL. To address them, we propose the 1) *Distributional La-*

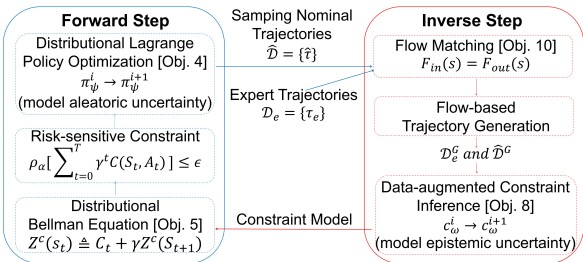

Figure 1: The flowchart of UAICRL.

*grange Policy Optimization (DLPO)* algorithm with a risk-sensitive constraint by modeling the distribution of the cumulative costs (Section 3), and 2) *Data-augmented Constraint Inference* with flow-based trajectory generation for constraint inference from limited demonstrations (Section 4). Figure 1 shows the flowchart of UAICRL framework.

## 3  POLICY OPTIMIZATION WITH RISK-SENSITIVE CONSTRAINTS

To address the aleatoric uncertainty, risk-sensitive RL commonly incorporates the risk measure into the optimization objective, for example, by constructing $\arg\max_\pi \rho_\alpha^\pi[\sum_{t=0}^T \gamma^t O(S_t, A_t)]$ where $O$ refers to the objective, $\alpha \in [0, 1]$ defines the confidence level and $\rho$ denotes the risk measure. This approach imposes significant costs on the policies that may lead to hazardous consequences, even if the likelihood of such events is low (e.g., a pedestrian being hit by a car on the highway). Although safety is important, being too conservative can harm performance and hinder reward accumulation (Mihatsch & Neuneier, 2002; Choi et al., 2021).

**Risk-Sensitive Constraints for Policy Optimization.** In this work, we present a novel approach that disentangles the unified objective into two components: rewards and costs, where rewards indicate the desired outcome, while costs account for negative consequences. For example, in autonomous driving, rewards might correspond to how quickly a car reaches its destination, while costs capture the need to adhere to traffic rules and safety constraints during driving. To drive a risk-sensitive policy, the risk incurred by aleatoric uncertainty can be incorporated separately from reward optimization. We formulate the trade-off between rewards and costs as a constrained optimization problem:

$$\arg\max_\pi \mathbb{E}_{\pi, p_\mathcal{T}, p_\mathcal{R}, \mu_0} \left[ \sum_{t=0}^T \left( \gamma^t r(s_t, a_t) + \beta\gamma^t \mathcal{H}[\pi(a_t|s_t)] \right) \right] \text{ s.t. } \rho_\alpha \left[ \sum_{t=0}^T \gamma^t C(S_t, A_t) \right] \le \epsilon \quad (4)$$

where $\mathcal{H}[\pi(a_t|s_t)]$ refers to the causal entropy (Ziebart et al., 2010) and $C(\cdot)$ denotes the random variable of state-action cost [1]. To specify the risk measure, we define the trajectory-generating probability as $p^\pi(\tau) = \mu_0(s_0) \prod_{t=0}^{T-1} \pi(a_t|s_t) p_\mathcal{T}(s_{t+1}|s_t, a_t)$, where aleatoric uncertainty accumulates in sequential decision-making due to the inherent stochasticity in the initial state distribution $\mu_0(s_0)$, policy $\pi$, and transition function $p_\mathcal{T}(s_{t+1}|s_t, a_t)$. We define the corresponding risk envelope $\mathcal{U}_\alpha^\pi = \{\zeta_\alpha : \Gamma \to [0, \frac{1}{\alpha}] | \sum_{\tau \in \Gamma} \zeta(\tau)p^\pi(\tau) = 1\}$ to be a compact, convex, and bounded set, based on which the risk measure can be induced by the distorted probability distribution $p_\zeta^\pi = \zeta \cdot p^\pi$. For example, the Conditional Value-at-Risk (CVaR) can be defined as $\rho_\alpha^\pi[\sum_{t=0}^T \gamma^t C_t] = \sup_{\zeta_\alpha \in \mathcal{U}_\alpha^\pi} \mathbb{E}_{\tau \sim p^\pi}[\zeta_\alpha(\tau) \sum_{t=0}^T \gamma^t C_t]$.

**Distributional Estimator for Cumulative Costs.** The difficulty of constructing the distorted probability-based risk measure lies in the unknown distribution of the cumulative costs. To estimate the distribution, we define the variable of discounted cumulative costs as $Z^c(s_t) = \sum_{\iota=0}^{T-t} \gamma^\iota C_\iota | S_0 = s_t$. Based on Luo et al. (2022), we use $N$ supporting quantiles parameterized with the rational-quadratic function to represent $Z_\theta^c$, where $\theta$ denotes model parameters. Then we can update $Z_\theta^c$ through quantile regression (Bellemare et al., 2017).

---

[1]Throughout this paper, we use uppercase letters (e.g., $C$) to represent random variables and lowercase letters (e.g., $c$) to represent instances of variables. For brevity, we use $C_t$ as a shorthand of $C(S_t, A_t)$.

During training, when the agent performs an action $a \sim \pi(\cdot|s)$ at state $s$, the agent receives a cost $c \sim p_{\mathcal{C}}(\cdot|s, a)$ and moves to a future state $s' \sim p_{\mathcal{T}}(\cdot|s, a)$. This stochastic process can be captured by the distributional Bellman equation (state-form) (Gerstenberg et al., 2023):

$$Z_\theta^c(s) = \int_{a \in \mathcal{A}} \pi(a|s) \int_{s' \in \mathcal{S}} p_{\mathcal{T}}(s'|s, a) \int_{c \in \mathcal{C}} p_{\mathcal{C}}(c|s, a)(b_{c,\gamma})_{\#} Z_\theta^c(s') \, \mathrm{d}a \, \mathrm{d}s' \, \mathrm{d}c \qquad (5)$$

where the pushfoward operation $(b_{c,\gamma})_{\#}$ involves shifting and scaling the distribution by $c$ and $\gamma$. For brevity, we denote the distributional Bellman equation as $Z^c(s) \overset{\Delta}{:=} C(s, A) + \gamma Z^c(S')$. We show that the predicted distribution is sensitive to the aleatoric uncertainty.

**Proposition 1.** *(Liu et al., 2022). The key components for representing the aleatoric uncertainty can be captured by the distributional Bellman equation under the measure of entropy.*

The detailed proof is in Appendix A.1. Leveraging the aforementioned policy optimization objective (4) and distributional estimator (5), we design the Distributional Lagrange Policy Optimization (DLPO) algorithm to learn the policy under risk-sensitive constraints. The implementation details are shown in Algorithm 2 in Appendix B.1.

## 4  CONSTRAINT INFERENCE WITH FLOW-BASED DATA AUGMENTATION

Epistemic uncertainty arises due to the limited training data and the model's lack of knowledge about Out-of-Distribution (OoD) data. A common measure of epistemic uncertainty is the mutual information $I(\omega; y|\boldsymbol{x}, \mathcal{D})$ (Smith & Gal, 2018; van Amersfoort et al., 2020), which quantifies the amount of information gained by the model $\omega$ when it observes the true label $y$ for a given input $\boldsymbol{x}$. The greater the uncertainty of the model regarding the data, the more additional information it can obtain once the true label $y$ is observed.

Intuitively, epistemic uncertainty arises when the constraint model $\omega$ is required to predict the cost of a trajectory $\bar{\tau}$ that locates OoD of training data (i.e., predict $c(\bar{\tau}) = 1 - p(\phi|\bar{\tau}; \omega)$), which is generated by exploratory behaviors during policy updates. To reduce the effect of epistemic uncertainty, we need to minimize the mutual information $I(\omega; \Phi|\bar{\tau}, \mathcal{D})$, which can be represented as follows:

**Proposition 2.** *Let $\mathcal{D}$ denote the training dataset consisting of expert trajectories $\{\tau_e\}$ and imitation trajectories $\{\hat{\tau}\}$. Let $\bar{\tau}$ denote an OoD trajectory. Let $\omega$ denote the constraint model parameters and $q(\omega)$ denote the dropout distribution. $I(\omega; \Phi|\bar{\tau}, \mathcal{D})$ can be empirically represented by:*

$$\mathcal{H}[p(\Phi|\bar{\tau}, \mathcal{D})] - \frac{1}{M} \sum_m \mathcal{H}[p(\Phi|\bar{\tau}; \omega_m)] \text{ where } \omega_m \sim q(\omega) \qquad (6)$$

The proof is in Appendix A.2. Specifically, 1) $\mathcal{H}[p(\Phi|\bar{\tau}, \mathcal{D})] \in [0, \infty)$ measures the amount of information required to describe the feasibility $\Phi$ of an exploratory trajectory $\bar{\tau}$ based on the given training dataset $\mathcal{D}$. 2) $\mathcal{H}[p(\Phi|\bar{\tau}, \omega_m)]$ defines the entropy of constraint model parameterized by $\omega_m$. To reduce the epistemic uncertainty, we integrate them into our ICI objective as follows.

### 4.1  DATA-AUGMENTED INVERSE CONSTRAINT INFERENCE

To reduce the impact of epistemic uncertainty, we can augment the conventional ICI objective (Obj. 3) with the mutual information term $I(\omega; \Phi|\bar{\tau}, \mathcal{D})$. Since $I(\omega; \Phi|\bar{\tau}, \mathcal{D})$ is intractable, based on Proposition 2, we instead maximize the following objective with its empirical representation:

$$(7)$$
$$\frac{1}{M} \sum_m \mathbb{E}_{\mathcal{D}_e} \Big[ \sum_{t=0}^{T} \log[p(\phi|s_t^e, a_t^e; \omega_m)] \Big] - \mathbb{E}_{\hat{\mathcal{D}}} \Big[ \sum_{t=0}^{T} \log[p(\phi|\hat{s}_t, \hat{a}_t; \omega_m)] \Big] - \alpha \Big( \mathcal{H}[p(\Phi|\bar{\tau}, \mathcal{D})] - \mathcal{H}[p(\Phi|\bar{\tau}; \omega_m)] \Big)$$

where $\alpha$ controls the trade-off between the log-likelihood and mutual information, and we use dropout layers (Srivastava et al., 2014) for $q(\omega)$ by following Smith & Gal (2018). Besides, the conditional entropy $\mathcal{H}[p(\Phi|\bar{\tau}, \mathcal{D})]$ is independent of $\omega$, which reaches zero (its minimum) when $p(\Phi, \bar{\tau}, \mathcal{D}) = p(\mathcal{D})$, indicating that the dataset has already recorded the trajectory and its feasibility (i.e., $(\Phi, \bar{\tau}) \in \mathcal{D}$). To achieve the goal, we expand the dataset by generating trajectories $\{(\Phi^G, \tau^G)\}$, obtaining the augmented expert and nominal dataset $\mathcal{D}_e^G$ and $\hat{\mathcal{D}}^G$. By substituting them to objective (7), we get the following data-augmented constraint inference objective:

$$\frac{1}{M} \sum_m \mathbb{E}_{\mathcal{D}_e^G} \Big[ \sum_{t=0}^{T} \log[p(\phi|s_t^e, a_t^e; \omega_m)] \Big] - \mathbb{E}_{\hat{\mathcal{D}}^G} \Big[ \sum_{t=0}^{T} \log[p(\phi|\hat{s}_t, \hat{a}_t; \omega_m)] \Big] + \alpha \mathcal{H}[p(\Phi|\bar{\tau}; \omega_m)] \qquad (8)$$

We propose an approach that can generate a diverse set of expert and nominal trajectories as follows.

## 4.2 FLOW-BASED TRAJECTORY GENERATION

We propose a Flow-based Trajectory Generation (FTG) algorithm to perform conditional generation for maximizing $p(\Phi, \bar{\tau}|\mathcal{D})$. Unlike the traditional trajectory generation with policy rollout, FTG learns the transition densities from the trajectory dataset $\mathcal{D}$. Moreover, the generation is guided by the feasibility $\Phi = \{\phi, \phi^-\}$ of trajectories, rather than the reward-maximizing policy.

To learn FTG, we define the non-negative trajectory flow function $F : \mathcal{T} \to \mathcal{R}^+$, where $\mathcal{T}$ denotes the trajectory set. Intuitively, $F(\tau)$ quantifies the mass of particles (Bengio et al., 2021) passing through $\tau$, and denser particles indicate a higher probability of generating the trajectory. Under this setting, the state flow $F(s)$ is the integral of trajectory flows passing through this state: $F(s_t) = \int_{\tau \ni s_t} F(\tau) \mathrm{d}\tau$. The transition probabilities can be defined as $p_F(s_{t+1}|s_t) = \frac{F(s_t \to s_{t+1})}{F(s_t)}$ where $F(s_t \to s_{t+1}) = \int_{\tau=(\ldots,s_t \to s_{t+1},\ldots)} F(\tau)$ embeds the flow passing through the action $a_t$. The trajectory generation probability $p_F(\tau) = \prod_{t=0}^{T-1} p_F(s_{t+1}|s_t)$. In order to generate $\{\Phi^G, \tau^G\}$, the generated trajectories $\tau$ must correspond to a specific $\phi$. This is achieved by a careful design of the reward function. The reward functions for generating the expert ($\Phi = \phi$) and nominal trajectories ($\Phi = \phi^-$) can be set as:

$$R_e(s_t) = \begin{cases} 1, & t = T \land s_t \in \tau_e \\ 0, & \text{otherwise} \end{cases} \quad \text{and} \quad \hat{R}(s_t) = \begin{cases} 1, & t = T \land s_t \in \hat{\tau} \\ 0, & \text{otherwise} \end{cases} \tag{9}$$

To generate expert trajectories, we assign a positive reward (+1) to the terminated state of the recorded expert trajectories $\tau_e \in \mathcal{D}_e$ and 0s to others. The same approach is applied for generating nominal trajectories. We train two flow functions $F_e$ and $\hat{F}$ by utilizing $R_e(s_t)$ and $\hat{R}(s_t)$ respectively.

**Learning Flow Functions.** A common approach to updating the flow function $F(\cdot)$ is flow matching: for a state $s_t \in \mathcal{S}$, the inflows $\int_{a:\mathcal{T}(s,a)=s_t} F(s,a)\mathrm{d}a$ must match outflows $\int_{a \in \mathcal{A}} F(s_t, a)\mathrm{d}a$. To enable flow matching in continuous environments, we sample $M$ actions independently and uniformly from the continuous action space $\mathcal{A}$. These actions serve as an approximation of flows that can be utilized to train the flow network. The resulting loss function is approximated as Li et al. (2023):

$$\tag{10}$$
$$\mathcal{L}_\xi(\tau) = \sum_{s_t=s_0}^{s_T} \left\{ \log\left[\epsilon + \sum_{m=1}^{M} \exp F_\xi\Big(G(s_t, a_m), a_m\Big)\right] - \log\left[\epsilon + \lambda R(s_t) + \sum_{m=1}^{M} \exp F_\xi(s_t, a_m)\right] \right\}^2$$

where: 1) $\xi$ denotes the parameters of the flow network $F(\cdot)$. 2) $G(\cdot)$ is a retrieval neural network with $(s_{t+1}, a_t)$ as the input and $s_t$ as the output (i.e., to predict the parent of a state), which can be trained by MSE Loss based on given trajectories. 3) $\sum_{m=1}^{M} F_\xi(s_t, a_m)$ is an approximation of flows with $M$ sampled actions. 4) $\epsilon$ can balance the use of small and large flows, thereby preventing numerical problems when computing the logarithm of extremely small flows.

**Trajectory Generation.** FTG aims to generate trajectories $\tau^G = (s_0, a_0, \ldots, s_T, a_T)$ by starting from the initial state $s_0$ and iteratively sampling the action $a_t$ based on the scale of the flow $F(s_t, a_t)$, thereby transitioning to the next state $s_{t+1}$. In this process, actions with larger flow will be sampled with higher probabilities. To label $\tau^G$, we set $\Phi$ to $\phi$ if the generation is based on the expert flows (i.e., $F_e$, learned with $R_e$), and $\phi^-$ for nominal ones. The detailed implementation of FTG is shown in Algorithm 3 in Appendix B.1. Leveraging the modeling capability of the generative model, FTG learns intricate feature patterns of input data, generating trajectories that resemble the training data in the latent space, even if they are absent in the original state-action space. The incorporation of this data significantly mitigates the impact of epistemic uncertainty during constraint inference.

Building upon the aforementioned ideas for modeling aleatoric and epistemic uncertainties, Algorithm 1 presents the whole UAICRL algorithm.

## 5 RELATED WORKS

**Learning constraints from demonstrations.** Given the demonstrations, prior works mainly focused on inferring implicit constraints by determining the permissibility of actions under specific states. In *discrete* state-action spaces, Chou et al. (2020b); Park et al. (2020) learned constraint sets to differentiate feasible from infeasible state-action pairs. Scobee & Sastry (2020) proposing inferring the constraint set under the maximum entropy principle, and McPherson et al. (2021); Baert et al. (2023) extended it to stochastic environments using maximum causal entropy (Ziebart et al., 2010). In *continuous* domains, the goal is to infer boundaries between feasible and infeasible state-action pairs.

---

**Algorithm 1:** Uncertainty-aware Inverse Constrained Reinforcement Learning (UAICRL)

---

**Input:** Expert dataset $\mathcal{D}_e = \{\tau_e\}$, augmented data size $I$.

Initialize the cost model $c_\omega$, policy $\pi_\psi$, expert flow $F_\xi^e$, and nominal flow $\hat{F}_{\bar{\xi}}$;

Update $F_\xi^e$ with $\mathcal{D}_e$ by minimizing the flow loss (10);

**do**

    Update $\pi_\psi$ based on the risk-sensitive CRL objective (4) and distributional estimator (5);

    Sample nominal trajectories $\hat{\mathcal{D}} = \{\hat{\tau}\}$ with $\pi_\psi$ in the environment, and update $\hat{F}_{\bar{\xi}}$ with $\hat{\mathcal{D}}$ by the flow loss (10);

    Initialize the augmented expert and nominal datasets $\mathcal{D}_e^G = \mathcal{D}_e$ and $\hat{\mathcal{D}}^G = \hat{\mathcal{D}}$;

    **for** $i = 1, 2, \ldots, I$ **do**

        Generate expert and nominal trajectories $\tau_e^G$ and $\hat{\tau}^G$ with $F_\xi^e$ and $\hat{F}_{\bar{\xi}}$;

        Add the generated trajectories to datasets: $\mathcal{D}_e^G = \mathcal{D}_e^G \cup \tau_e^G$ and $\hat{\mathcal{D}}^G = \hat{\mathcal{D}}^G \cup \hat{\tau}^G$;

    **end**

    Update the cost model $c_\omega$ with the ICI objective (8) based on $\mathcal{D}_e^G$ and $\hat{\mathcal{D}}^G$;

**while** $\hat{\mathcal{D}}$ *do not match* $\mathcal{D}_e$;

---

Malik et al. (2021); Gaurav et al. (2023); Qiao et al. (2023) used neural networks to approximate constraints. Some recent studies (Liu et al., 2023; Chou et al., 2020a; Papadimitriou et al., 2023) applied Bayesian Monte Carlo and variational inference to infer a posterior distribution of constraints in high-dimensional state space. These constraint distributions can only model the epistemic uncertainty.

**Uncertainty-Aware Reinforcement Learning.** Incorporating uncertainty awareness in RL algorithms is essential for efficient exploration and control, with numerous downstream applications (Hoel et al., 2023; Liu et al., 2022; Wu et al., 2023). Several existing works study defining uncertainty measures in RL environments, primarily using dropout layers (Chen et al., 2017) or ensemble models (Lütjens et al., 2019; An et al., 2021), such as Bootstrapped DQN methods (Osband et al., 2016; Da Silva et al., 2020) and their offline extensions (Kumar et al., 2019; An et al., 2021). Another approach to measuring uncertainty is through Distributional RL (Luo et al., 2022; Bellemare et al., 2017; Dabney et al., 2018; Mavrin et al., 2019), which directly models the distribution of future returns with distributional Bellman operator. Besides, Risk-sensitive RL (Mihatsch & Neuneier, 2002; Chow et al., 2015; Prashanth et al., 2022; Ni & Lai, 2022; Luo et al., 2023) is indeed a closely related research area, in which agents employ risk measures, such as exponential utility, variance, and Value-at-Risk, to develop risk-aware policies. In contrast to prior works that apply risk measures for rewards maximization, our work builds upon CMDPs and emphasizes capturing the cost uncertainty.

## 6 EMPIRICAL EVALUATION

In this section, we empirically evaluate the effectiveness of UAICRL by answering the following questions: 1) How well does UAICRL perform in discrete (Section 6.1) and continuous (Section 6.2) environments under varying degrees of stochasticity? 2) Can UAICRL handle more complex, real-world highway driving scenarios (Section 6.3)? 3) How well do the key components of UAICRL (DLPO and FTG) address the aleatoric and epistemic uncertainty, respectively (Section 6.4)?

**Comparison Methods.** Our experiments include the following baselines: 1) *Generative Adversarial Constraint Learning* (**GACL**) extends Generative Adversarial Imitation Learning (GAIL) (Ho & Ermon, 2016) and employs a discriminator $D(s, a)$ to differentiate validate state-action pairs from infeasible ones. The $log D(s, a)$

Table 1: Baseline methods for constraint inference.

| Method | Continues Space | Constraint Optimization | Maximum Entropy | Aleatoric Uncertainty | Epistemic Uncertainty |
|---|---|---|---|---|---|
| GACL | ✓ | ✗ | ✗ | ✗ | ✗ |
| B2CL | ✓ | ✓ | ✗ | ✗ | ✗ |
| ICRL | ✓ | ✓ | ✓ | ✗ | ✗ |
| VICRL | ✓ | ✓ | ✓ | ✗ | ✓ |
| UAICRL-NRS | ✓ | ✓ | ✓ | ✗ | ✓ |
| UAICRL-NDA | ✓ | ✓ | ✓ | ✓ | ✗ |
| UAICRL | ✓ | ✓ | ✓ | ✓ | ✓ |

is directly appended to the reward function as a penalization. 2) *Binary Classifier Constraint Learning* (**BC2L**) employs a binary classifier as the constraint model without utilizing the Maximum Entropy (MEnt) framework. 3) *Inverse Constrained Reinforcement Learning* (**ICRL**) (Malik et al., 2021)

follows the MEnt framework with a risk-neutral constraint. 4) *Variational ICRL* (**VICRL**) (Liu et al., 2023) captures epistemic uncertainty in constraint inference using Beta distribution based on ICRL. Additionally, we perform ablation studies where 5) **UAICRL-NRS** removes the risk-sensitive constraint from UAICRL and 6) **UAICRL-NDA** removes the data augmentation from UAICRL. Table 1 summarizes these methods.

**Experiment Settings.** We conduct empirical evaluations utilizing an ICRL benchmark (Liu et al., 2023), and extend it to include stochastic dynamics by incorporating noise into transitions. Following Malik et al. (2021), evaluation metrics include: 1) *Constraint Violation Rate* measures the probability that a policy violates constraint in a trajectory. 2) *Feasible Cumulative Rewards* calculate the total rewards obtained by the agent before violating any constraints.

## 6.1 DISCRETE ENVIRONMENT: STOCHASTIC GRIDWORLD

We construct three Gridworld environments with different constraints, where the agent's objective is to navigate from a starting location to a target location while avoiding the added constraints, as shown in Figure 2. The environment exhibits a certain degree of stochasticity, where, with a specific probability ($p_s = 0.01$ and $0.001$), the environment receives a random action instead of an agent's action. To be compatible with the environment set-

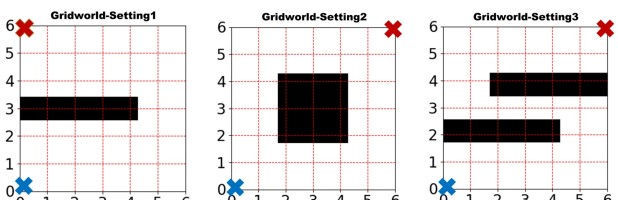

Figure 2: Three different Gridworld settings with blue, red, and black markers indicating the starting, target, and constrained locations respectively.

tings, we utilize the discretized implementation of UAICRL by following Liu et al. (2023); Malkin et al. (2022). Appendix B.2 shows the implementation details.

Table 2 shows the evaluation performance, where bolding denotes the best results (highest rewards or lowest violation rates) in each setting. Check Appendix D.1 for training curves. We find that UAICRL generally outperforms other methods with lower constraint violation rates and higher rewards, while GACL and ICRL can hardly work well. BC2L and VICRL achieve relatively satisfactory results in the first two settings but perform poorly in the third setting.

Table 2: Evaluation of different methods in three Gridworlds with the random rate $p_s = 0.01$ and $0.001$.

| | | $p_s = 0.01$ | | | $p_s = 0.001$ | | |
|---|---|---|---|---|---|---|---|
| | | Gridworld Setting 1 | Gridworld Setting 2 | Gridworld Setting 3 | Gridworld Setting 1 | Gridworld Setting 2 | Gridworld Setting 3 |
| Feasible Rewards | BC2L | 0.451 | **0.716** | 0.125 | 0.647 | 0.602 | 0.192 |
| | GACL | 0.032 | 0.109 | 0.000 | 0.011 | 0.070 | 0.000 |
| | ICRL | 0.244 | 0.532 | 0.033 | 0.356 | 0.368 | 0.089 |
| | VICRL | 0.537 | 0.310 | 0.051 | 0.778 | 0.610 | 0.070 |
| | UAICRL | **0.650** | 0.683 | **0.359** | **0.797** | **0.739** | **0.401** |
| Constraint Violation Rate | BC2L | 0.33 | 0.19 | 0.58 | 0.29 | 0.27 | 0.52 |
| | GACL | 0.43 | 0.29 | 0.78 | 0.67 | 0.11 | 0.84 |
| | ICRL | 0.53 | 0.33 | 0.63 | 0.36 | 0.27 | 0.73 |
| | VICRL | 0.35 | 0.33 | 0.45 | 0.19 | 0.28 | 0.53 |
| | UAICRL | **0.13** | **0.09** | **0.34** | **0.09** | **0.07** | **0.38** |

## 6.2 CONTINUOUS ENVIRONMENT: STOCHASTIC MUJOCO

We utilize five MuJoCo environments in the benchmark (Liu et al., 2023) and additionally incorporate Gaussian noise into the transition function as $p_{\mathcal{T}}(s_{t+1}|s_t, a_t) = f(s_t, a_t) + \mathcal{N}(\mu, \sigma)$. Each experiment is repeated with four random seeds, over which the mean ± standard deviation (std) results are reported. More details are shown in Appendix C.2.

Figure 3 displays the training curves in stochastic MuJoCo environments with noise $\sigma = 0.1$ (check Appendix D.2 for results with $\sigma = 0.01$ and $0.001$). We observe that UAICRL-NRS and UAICRL-NDA can generally outperform the baselines in most environments, which underscores the effectiveness of employing data augmentation for modeling epistemic uncertainty and utilizing distributional estimator for capturing aleatoric uncertainty. By modeling both uncertainties, UAICRL generally obtains high feasible rewards (bottom row Fig. 3) whilst having a low constraint violation rate (top row Fig. 3). This demonstrates UAICRL's robustness and its superior performance compared to other methods. Note that although the reward shaping method, GACL, exhibits the fewest constraint violations in the Blocked Ant environment, it struggles to optimize rewards and underperforms in other environments. This underperformance is primarily attributed to the estimation errors encountered during the early stages of training. Interestingly, we find that increased stochasticity does not necessarily result in poorer model performance. This is because larger noise levels are more readily detectable, leading models to exhibit sensitivity to the risk of constraint violations. For a more comprehensive analysis of these phenomena, we provide a more detailed discussion in Appendix E.

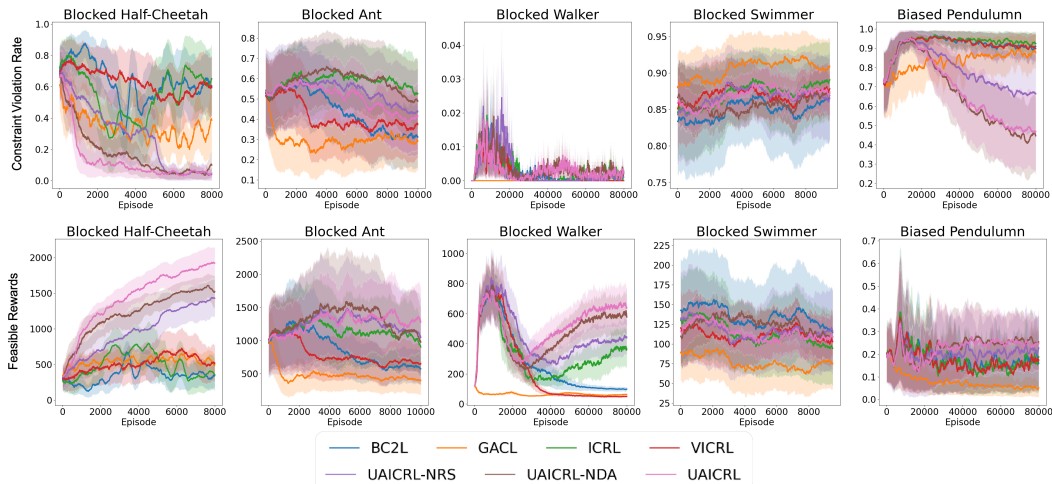

Figure 3: The constraint violation rate (top) and feasible rewards (bottom) in five MuJoCo environments during training with stochasticity of $\mathcal{N}(0, 0.1)$.

### 6.3 REALISTIC ENVIRONMENT: STOCHASTIC HIGHWAY DRIVING

We conduct experiments in a realistic high-dimensional Highway Driving (HighD) environment (Krajewski et al., 2018; Liu et al., 2023), which defines a complex highway driving task with constraints and requires the agent to drive safely to the destination by observing human drivers' demonstrations. In this paper, we focus on the velocity constraint, which guarantees the ego car to drive at a safe velocity. Additionally, we add Gaussian noise to the agent's action $a_t$ to introduce the control-level stochasticity. We report both the constraint violation rate (left, Figure 4) and feasible rewards (right, Figure 4) under noise $\mathcal{N}(0, 0.1)$ (check Appendix D.3 for complete results with $\mathcal{N}(0, 0.01)$ and $\mathcal{N}(0, 0.001)$).

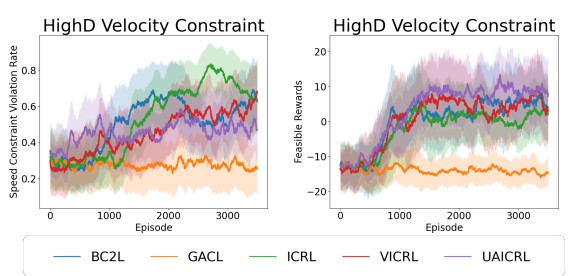

Figure 4: Model performance in the HighD environment with velocity constraint under $\mathcal{N}(0, 0.1)$.

Additionally, Appendix D.4 shows the results of recovering multiple kinds of constraints. We find that although GACL violates the fewest constraints, it is too conservative to pursue rewards, which limits its practical application. Other baselines including BC2L, ICRL, and VICRL can obtain high rewards, but their constraint violation rates are still relatively high. Instead, UAICRL can achieve the highest rewards while maintaining a satisfactorily low constraint violation rate, indicating its great potential to handle uncertainties even in complex environments.

### 6.4 IN-DEPTH STUDY ON UNCERTAINTY MANAGEMENT PERFORMANCE

To comprehend the critical factors contributing to uncertainty awareness, we study the model's capabilities of representing aleatoric and epistemic uncertainties from the following perspectives:

**"Can DLPO lead to more robust control under stochastic dynamics?"** To address this question, we provide each agent with the ground-truth constraints in the environment. We focus solely on whether DLPO can learn constraint-satisfying behavior while being influenced by aleatoric uncertainty. We report both the constraint violation rate (left, Fig. 5) and feasible rewards (right, Fig. 5) in the Half-Cheetah environment with stochasticity of $\mathcal{N}(0, 0.1)$ (check Appendix D.2 for complete results). The compared methods include: 1) *DLPO-Neutral* and *DLPO-Averse*, which refer to the options of applying expectation and CVaR(0.25) as the risk measures ($\rho$) in Objective 4. 2) *Classic PPO-Lagrange*,

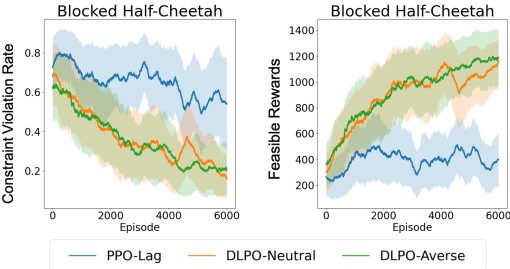

Figure 5: Model performance in the Blocked Half-Cheetah environment with stochasticity of $\mathcal{N}(0, 0.1)$ when given ground-truth constraints.

following the implementation in Liu et al. (2023). Our findings indicate that by implementing the risk-sensitive constraint, the policy learns to respect the constraints while collecting rewards. Furthermore, in most cases, the DLPO-Averse method employing CVaR as the risk measure results in fewer constraint violations compared to the risk-neutral approach. However, CVaR sometimes leads to overly conservative behavior, hindering the agent from efficiently pursuing rewards.

Based on the same motivation, we train agents in Gridworld environments under known constraints to better understand the advantages of the distributional estimator. Fig. 6 visualizes the trajectories generated by the PPO-Lagrange (left) and DLPO (right) in a Girdworld setting (check Fig. D.3 for results in remaining settings). We notice that DLPO can maintain an appropriate distance from

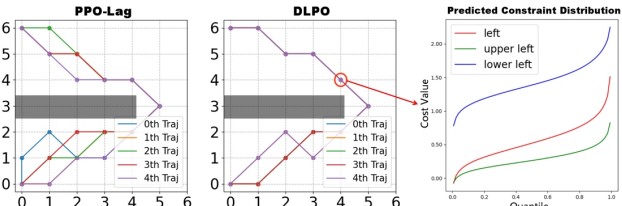

Figure 6: The trajectories generated by PPO-Lag and DLPO, along with the predicted cost distributions at the critical state denoted by the red circle.

the constrained locations, effectively mitigating the impact of noise. Specifically, at $s(4, 4)$, PPO-Lag chooses to move left, whereas DLPO decides to move upper left, maintaining a larger gap to the constraints. We use red circles to highlight the critical locations, where the agents display different actions, and visualize the predicted cost distribution at these locations. We find that these distributions are sensitive to the choices of actions, consequently assigning larger expectations and variances to actions with a higher risk of constraint violation (e.g., moving lower left at $s(4, 4)$ in Fig. 6).

**"Can FTG facilitate more accurate constraint inference?"** Despite the ablation studies in the main experiments (e.g., compare UAICRL and UAICRL-NDA), for a better understanding of the effectiveness of FTG, we visualize the inferred constraint from MEICRL (Malik et al., 2021) and our UAICRL with FTG. Figure 7 illustrates the recovered costs in a

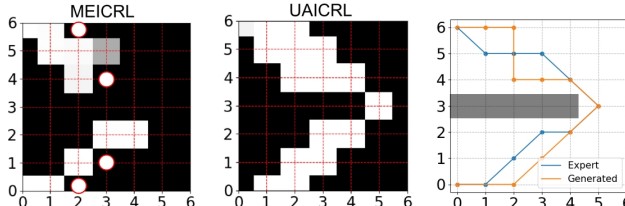

Figure 7: The constraint map recovered by MEICRL and UAICRL, along with the trajectories generated by our FTG.

Gridworld setting (check Fig. D.4 for results in remaining settings). The generated trajectory is not included in the expert dataset due to its restricted size. Generating such trajectories enhances the diversity of the dataset, mitigating the redundant constraints learned by MEICRL (refer to white circles). This finding suggests that when trained with additionally generated trajectories, the constraint model can predict step-wise costs more accurately, thereby reducing the epistemic uncertainty.

### 6.5 LIMITATIONS

**Theoretical Results.** While our study indeed encompasses theoretical findings, it still lacks a rigorous theoretical analysis. Existing theoretical results on IRL (Metelli et al., 2021; Lindner et al., 2022; Metelli et al., 2023) depended on the construction of the feasible set of rewards, but defining the exact feasible set for the constraint is challenging due to its various types (e.g., hard, soft or probabilistic constraints) and optimization techniques (e.g., Lagrange methods). A rigorous theoretical study is beyond the scope of our current work. Nevertheless, we affirm the importance of such understanding and suggest extending the theoretical outcomes from IRL to ICRL as a future work.

**Limited Risk Measure.** For fairness and simplicity, our study predominantly employs the widely used CVaR method. Although the performance of other metrics like VaR and Entropic Value-at-Risk (EVaR) has not been studied, they could be easily integrated into our framework for future exploration.

### 7 CONCLUSION

This paper presents UAICRL, a novel ICRL framework capable of modeling both aleatoric and epistemic uncertainties. This is achieved through the use of a distributional estimator and a flow-based generator, enabling uncertainty-aware constraint inference. Empirical results show that UAICRL outperforms a range of ICRL methods in both discrete and continuous environments, as well as in a real-world highway driving task. Looking forward, a promising extension would be to incorporate robust optimization methods into our UAICRL framework. This would enable learning more robust constraints for safe control in environments with unpredictable noise.

ACKNOWLEDGMENTS

The work is partly supported by Shenzhen Fundamental Research Program (General Program) under grant JCYJ20230807114202005, by the National Key R&D Program of China under grant No2022ZD0116004, by Shenzhen Science and Technology Program ZDSYS20211021111415025, and by Guangdong Provincial Key Laboratory of Mathematical Foundations for Artificial Intelligence.

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

## A  PROOF

### A.1  PROOF OF PROPOSITION 1

Let $\boldsymbol{Z}, \bar{\boldsymbol{C}}^\pi$ denote the vector-valued random variables of size $|\mathcal{S}|$, where $Z(s) = \sum_{t=0}^\infty \gamma^t C_t | S_0 = s$ and $\bar{C}^\pi(s) = \int_{a \in \mathcal{A}} \pi(a|s) C(s,a) \mathrm{d}a$. Let $\boldsymbol{P}^\pi$ denote the state transition probability matrix under policy $\pi$, where $P_{s,s'}^\pi = \int_{a \in \mathcal{A}} P(s'|s,a) \, \mathrm{d}\pi(a|s)$. Assuming the Bellman consistency holds by $\boldsymbol{Z} \overset{\Delta}{:=} \bar{\boldsymbol{C}}^\pi + \gamma \boldsymbol{P}^\pi \boldsymbol{Z}$, we show that the uncertainty of distributions $\boldsymbol{Z}$ under an entropy measure can be given by:

$$\mathcal{H}(\boldsymbol{Z}) = \mathcal{H}[\bar{\boldsymbol{C}}^\pi] - |\mathcal{S}| \log(1-\gamma) + \log|\det(\mathbf{d}^\pi)| \tag{11}$$

where $\mathbf{d}^\pi = (1-\gamma)(I - \gamma \boldsymbol{P}^\pi)^{-1} \in [0,1]^{|S| \times |S|}$ is the induced matrix for distributions over states by following policy $\pi$ and transition $P_{\mathcal{T}}$.

*Proof:*

$$H(\boldsymbol{Z}) \overset{(a)}{=} H\left[(I - \gamma \boldsymbol{P}^\pi)^{-1} \bar{\boldsymbol{C}}^\pi\right] \tag{12}$$

$$\overset{(b)}{=} \log\left|\det\left[(I - \gamma \boldsymbol{P}^\pi)^{-1}\right]\right| + H[\bar{\boldsymbol{C}}^\pi] \tag{13}$$

$$\overset{(c)}{=} \log\left|\det\left[\frac{\mathbf{d}^\pi}{1-\gamma}\right]\right| + H[\bar{\boldsymbol{C}}^\pi] \tag{14}$$

$$\overset{(d)}{=} -|S| \log(1-\gamma) + \log|\det[\mathbf{d}^\pi]| + H[\bar{\boldsymbol{C}}^\pi] \tag{15}$$

**(a)** holds by following the Bellman consistency $\boldsymbol{Z} \overset{\Delta}{:=} \bar{\boldsymbol{C}}^\pi + \gamma \boldsymbol{P}^\pi \boldsymbol{Z}$. To show the invertibility of $(I - \gamma \boldsymbol{P}^\pi)$, it is sufficient to demonstrate that the matrix is full rank. Specifically, for any non-zero vector $\boldsymbol{x} \in \mathbb{R}^{|S|}$:

$$\begin{aligned}
\|(I - \gamma \boldsymbol{P}^\pi)\boldsymbol{x}\|_\infty &= \|\boldsymbol{x} - \gamma \boldsymbol{P}^\pi \boldsymbol{x}\|_\infty \\
&\geq \|\boldsymbol{x}\|_\infty - \gamma \|\boldsymbol{P}^\pi \boldsymbol{x}\|_\infty \\
&\geq \|\boldsymbol{x}\|_\infty - \gamma \|\boldsymbol{x}\|_\infty \\
&= (1-\gamma)\|\boldsymbol{x}\|_\infty \\
&\geq 0
\end{aligned}$$

**(b)** holds by applying the properties of differential entropy and utilizing the invertibility of $(I - \gamma \boldsymbol{P}^\pi)$.

**(c)** holds by defining $\mathbf{d}^\pi = (1-\gamma)(I - \gamma \boldsymbol{P}^\pi)^{-1}$. We aim to demonstrate that $\mathbf{d}^\pi \in [0,1]^{|S| \times |S|}$ represents the induced matrix for distributions over states when following policy $\pi$. Specifically, the $(s)^{th}$ row of $\mathbf{d}^\pi$ corresponds to the distribution over states induced by following policy $\pi$ after starting with $s_0 = s$. This follows directly from the definition of $\mathbf{d}^\pi$.

$$\begin{aligned}
\mathbf{d}^\pi &= (1-\gamma) \sum_{t=1}^\infty (\gamma \boldsymbol{P}^\pi)^t \\
&= \frac{(1-\gamma)[1 - (\gamma \boldsymbol{P}^\pi)^\infty]}{1 - (\gamma \boldsymbol{P}^\pi)} \\
&= \frac{(1-\gamma)}{1 - (\gamma \boldsymbol{P}^\pi)}
\end{aligned}$$

Proposition 1 provides a decomposition of the entropy of $\boldsymbol{Z}$ into three components. 1) The first component is the entropy of the cost variables, which quantifies the uncertainty associated with the current costs. 2) The second component captures the uncertainty induced by the discount factor, which determines the degree to which the current uncertainty estimation should be influenced by the stochasticity of future rewards or transitions. 3) The third component is a log-absolute determinant of the induced distribution matrix, which measures the extent to which the transition function $P_{\mathcal{T}}$ and the policy $\pi$ stretch or alter the initial state-action distribution. These components correspond to the key elements of aleatoric uncertainty.

## A.2 PROOF OF PROPOSITION 2

*Proof:*

The mutual information term can be factorized as follows:

$$I(\omega; \Phi|\bar{\tau}, \mathcal{D}) \overset{(a)}{=} \mathcal{H}[p(\Phi|\bar{\tau}, \mathcal{D})] - \mathbb{E}_{p(\omega|\mathcal{D})}\Big[\mathcal{H}[p(\Phi|\bar{\tau}; \omega)]\Big] \tag{16}$$

$$\overset{(b)}{=} \mathcal{H}[p(\Phi|\bar{\tau}, \mathcal{D})] - \int p(\omega|\mathcal{D})\mathcal{H}[p(\Phi|\bar{\tau}; \omega)]\mathrm{d}\omega \tag{17}$$

$$\overset{(c)}{\simeq} \mathcal{H}[p(\Phi|\bar{\tau}, \mathcal{D})] - \int q(\omega)\mathcal{H}[p(\Phi|\bar{\tau}; \omega)]\mathrm{d}\omega \tag{18}$$

$$\overset{(d)}{\simeq} \mathcal{H}[p(\Phi|\bar{\tau}, \mathcal{D})] - \frac{1}{M}\sum_m \mathcal{H}[p(\Phi|\bar{\tau}; \omega_m)] \text{ where } \omega_m \sim q(\omega) \tag{19}$$

**(a)** holds by following the meaning of mutual information (i.e., the amount of information gained by the model $\omega$ if receiving a label $\Phi$ for a new trajectory $\bar{\tau}$, given the dataset $\mathcal{D}$).

**(c)** holds by using the variational inference to approximate the intractable posterior $p(\omega|\mathcal{D})$ with a simpler approximating distribution $q(\omega)$. For neural networks, the dropout distribution is commonly used.

**(d)** holds by sampling from the approximating distribution (e.g., the dropout distribution) with Monte Carlo method.

## B  IMPLEMENTATION DETAILS

### B.1  MORE ALGORITHMS

We show the Distributional Lagrange Policy Optimization (DLPO) and Flow-based Trajectory Generation (FTG) in Algorithm 2 and Algorithm 3, respectively.

---

**Algorithm 2:** Distributional Lagrange Policy Optimization (DLPO)

---

**Input:** Lagrange multiplier $\lambda$, risk measure $\rho$, GAE lambda $\lambda_g$, rollout rounds $B$, update rounds $\mathcal{K}$, policy $\pi_\psi$, reward value critic $V^r$ and distributional cost value critic $Z^c$;

Initialize state $s_0$ from CMDP and the roll-out dataset $\mathcal{D}_{roll}$;

**for** $b = 1, 2, \ldots, B$ **do**

    Perform policy $\pi_\psi$ and collect trajectories $\tau_b = [s_0, a_0, r_0, c_0, \ldots, s_T, a_T, r_T, c_T]$;

    Calculate reward advantages $A_t^r$ and return $R_t^r$ via GAE (Schulman et al., 2016) ;

    Calculate cost advantages $A_t^c = \sum_{\iota=t}^T (\gamma\lambda_g)^\iota [c_\iota + \gamma\rho(Z^c(s_{\iota+1})) - \rho(Z^c(s_\iota))]$;

    Add samples to the dataset $\mathcal{D}_{roll} = \mathcal{D}_{roll} \cup \{s_t, a_t, r_t, A_t^r, R_t^r, c_t, A_t^c\}$;

**end**

**for** $\kappa = 1, 2, \ldots, \mathcal{K}$ **do**

    Sample a data point $s_\kappa, a_\kappa, r_\kappa, A_\kappa^r, R_\kappa^r, c_\kappa, A_\kappa^c$ from the dataset $\mathcal{D}_{roll}$;

    Calculate the clipping loss $L^{CLIP} =$

    $\min\left[\frac{\pi_\psi(a_\kappa|s_\kappa)}{\pi_{\psi,old}(a_\kappa|s_\kappa)}(A_\kappa^r - \lambda(A_\kappa^c - \epsilon)), clip(\frac{\pi_\psi(a_\kappa|s_\kappa)}{\pi_{\psi,old}(a_\kappa|s_\kappa)}, 1 - \delta, 1 + \delta)(A_\kappa^r - \lambda(A_\kappa^c - \epsilon))\right]$;

    Update policy parameters $\psi$ by minimizing the loss: $-L^{CLIP} - \beta\mathcal{H}[\pi_\psi(a_\kappa|s_\kappa)]$ ;

    Update the reward critic $V^r$ by minimizing the loss: $L^{VF} = \|V^r(s_\kappa) - R_\kappa\|_2^2$;

    Update the cost distribution $Z^c$ by distributional Bellman operator (Equation 5) ;

**end**

Update the Lagrange multiplier $\lambda$ by minimizing the loss $L^\lambda = \lambda[\mathbb{E}_{\mathcal{D}_{roll}}(c) - \epsilon]$ ;

---

### B.2  DISCRETIZED IMPLEMENTATION

We follow the Policy Iteration Lagrange algorithm (see Algorithm 2 in Liu et al. (2023)) to solve discretized control problems. Based on it, we simply incorporate a distributional term into the value

---

**Algorithm 3:** Flow-based Trajectory Generation (FTG)

---

**Input:** Flow neural network $F_\xi$, retrieval neural network $G_\chi$, trajectory dataset $\mathcal{D} = \{\tau_i\}_{i=1}^N$,
   empty probability buffer $\mathcal{P}$, empty generated trajectory buffer $\mathcal{D}^G$, generate rounds $I$;

// *Flow matching*

**while** *not converge* **do**

   Sample a minibatch $\mathcal{B}$ of trajectory data from $\mathcal{D}$;

   Update retrieval network $G_\chi$ by minimizing the loss: $MSELoss(s_t, G_\chi(s_{t+1}, a_t))$;

   Uniformly sample $M$ actions $\{a_m\}_{m=1}^M$ from action space $\mathcal{A}$ for each state $s_t$ in $\mathcal{B}$;

   Calculate inflows $F_{in} = \log\left[\epsilon + \sum_{m=1}^M \exp F_\xi\left(G_\chi(s_t, a_m), a_m\right)\right]$;

   Calculate outflows $F_{out} = \log\left[\epsilon + \lambda R(s_t) + \sum_{m=1}^M \exp F_\xi(s_t, a_m)\right]$;

   Update flow network $F_\xi$ by flow matching (Eqn. 10);

**end**

// *Trajectory generation*

**for** $\kappa = 1, 2, \dots, I$ **do**

   Initialize state $s_0$ from CMDP, set $t = 0$;

   Append $s_0$ to the generated trajectory $\tau_\kappa^G$;

   **while** $s \neq terminal$ **do**

      Uniformly sample $K$ actions $\{a_i\}_{i=1}^K$ from action space $\mathcal{A}$;

      Generate the action probability buffer $\mathcal{P}$ by calculating edge flow for each action:

      $\mathcal{P} = \{F_\xi(s_t, a_i)\}_{i=1}^K$;

      Sample $a_t \sim \mathcal{P}$ and execute $a_t$ to move to the next state $s_{t+1}$;

      Append $s_{t+1}$ to the generated trajectory $\tau_\kappa^G$;

   **end**

   Append generated trajectory $\tau_\kappa^G$ to buffer $\mathcal{D}^G$;

**end**

---

matrix and modify the flow-based data augmentation into a discretized version following Malkin et al. (2022).

### B.3    EXPERIMENTAL SETTINGS

For training the ICRL models, we utilized a total of 8 NVIDIA GeForce RTX 3090 GPUs, each equipped with 24 GB of memory. The training process was conducted on a single running node, utilizing 8 CPUs per task. The random seeds in the MuJoCo and HighD environment are 123, 321, 456, and 654. With the allocated resources described above, running one seed in the virtual and realistic environments typically required a duration of 2-4 and 3-5 hours, respectively. To optimize all of our networks, we employed the Adam optimization algorithm (Kingma & Ba, 2015). The learning rate was updated using an exponential decay schedule parameterized by a decay factor in every iteration. Given our primary focus on stochastic environments where absolute constraint satisfaction is challenging, we opt for a small but positive constraint bound $\epsilon = $1e-8. We summarize the main hyperparameters in Teble B.1.

## C    ENVIRONMENTAL DETAILS

### C.1    GRIDWORLD

In this paper, we establish a map with dimensions of $7 \times 7$ units and construct three distinct settings, as illustrated in Figure 2. At each time, the agent is permitted to navigate to any of the adjacent eight grids by moving one step. Starting from the initial location, a reward of 1 is granted if the agent successfully traverses to the target location while avoiding the imposed constraints, and a reward of 0 is assigned for any other situations. Moreover, we modify the environment to simulate a certain degree of stochasticity. Specifically, there exists a random rate ($p_s = 0.01$ and 0.001) with which the environment receives a random action instead of the intended action executed by the agent.

Table B.1: List of the utilized hyperparameters in UAICRL. To ensure equitable comparisons, we maintain consistency in the parameters of the same neural networks across different models.

| Parameters | HalfCheetah | Ant | Walker | Swimmer | Pendulum |
|---|---|---|---|---|---|
| General | | | | | |
| Expert Rollouts | 10 | 50 | 50 | 50 | 50 |
| Max Length | 1000 | 500 | 500 | 500 | 100 |
| Gamma | 0.99 | 0.99 | 0.99 | 0.99 | 0.99 |
| PPO | | | | | |
| Steps | 2048 | 2048 | 2048 | 2048 | 2048 |
| Reward-GAE-$\lambda$ | 0.95 | 0.9 | 0.9 | 0.9 | 0.9 |
| Cost-GAE-$\lambda$ | 0.95 | 0.9 | 0.9 | 0.95 | 0.9 |
| Policy Network $\pi_\theta$ | 64, 64 | 64, 64 | 64, 64 | 64, 64 | 64, 64 |
| Reward Network $V_\theta^r$ | 64, 64 | 64, 64 | 64, 64 | 64, 64 | 64, 64 |
| Cost Network $Z_\theta^c$ | 256, 256 | 256, 256 | 256, 256 | 256, 256 | 256, 256 |
| Learning Rate $\theta$ | 3e-4 | 3e-5 | 1e-4 | 3e-4 | 1e-4 |
| Target KL | 0.01 | 0.02 | 0.02 | 0.01 | 0.02 |
| Lagrangian | | | | | |
| Initial Value | 1 | 0.05 | 0.1 | 1 | 0.1 |
| Learning Rate | 0.1 | 0.02 | 0.05 | 0.01 | 0.05 |
| Constraint Function | | | | | |
| Network $C_\omega$ | 20 | 40, 40 | 64, 64 | 20 | 20 |
| Learning Rate $\omega$ | 0.05 | 0.005 | 0.001 | 0.001 | 0.001 |
| Backward Iterations | 10 | 5 | 5 | 5 | 5 |
| DLPO | | | | | |
| Quantiles | 64 | 64 | 64 | 64 | 64 |
| Risk Measure | CVaR | CVaR | CVaR | CVaR | CVaR |
| Risk Level | 0.25 | 0.95 | 0.95 | 0.95 | 0.25 |
| FTG | | | | | |
| Network $F_\xi$ | 256, 256 | 256, 256 | 256, 256 | 256, 256 | 256, 256 |
| Learning Rate $\xi$ | 3e-4 | 3e-4 | 3e-4 | 3e-4 | 3e-4 |
| Sample Action Size | 10000 | 10000 | 10000 | 10000 | 10000 |
| Sample Flows | 50 | 50 | 50 | 50 | 50 |

## C.2 MuJoCo

The five MuJoCo robotics environments are built upon MuJoCo (see Figure C.1).

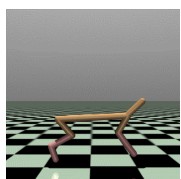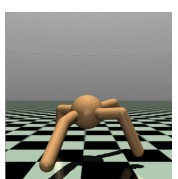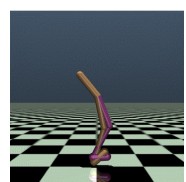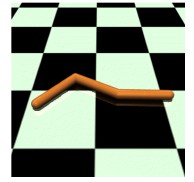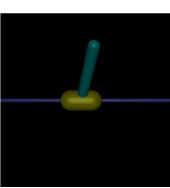

Figure C.1: MuJoCo environments. From left to right, the environments are Half-cheetah, Ant, Walker, Swimmer, and Inverted Pendulum, respectively.

We modify them by incorporating predefined constraints and adjusting reward terms for ICRL. To simulate the stochasticity in the environment dynamics, we incorporate a Gaussian noise $\eta \sim \mathcal{N}(\mu, \sigma)$ into the transition function at each step such that $p\left(s_{t+1} \mid s_t, a_t\right) = f\left(s_t, a_t\right) + \eta$. In this work, we utilize $\mu = 0$ with $\sigma = 0.1, 0.01$ and $0.001$ to represent different scales of noise. We provide a more comprehensive description as follows:

1) *Blocked Half-Cheetah.* In this environment, the agent controls a robot with two legs. The reward obtained by the agent is determined by the distance the robot walks between two time steps, and a penalty based on the magnitude of the input action. The game continues until a maximum time step of 1000 is reached. Due to the fact that the robot is easier to move backward than forward, we define a blocked region where the X-coordinate $< -3$. So the robot is restricted to move within the region where the X-coordinate $\geq -3$.

2) *Blocked Ant.* In this environment, the agent controls a robot with four legs. The rewards obtained by the agent depend on the distance of the robot from the origin, and a healthy bonus for maintaining

Table C.1: The stochastic MuJoCo environments with constraints

| Name | Obs. Dim. | Act. Dim. | Constraints | Stochasticity |
|------|-----------|-----------|-------------|---------------|
| Blocked Half-Cheetah | 18 | 6 | X-Coordinate $\geq$ -3 | $\mathcal{N}(\mu, \sigma)$ |
| Blocked Ant | 113 | 8 | X-Coordinate $\geq$ -3 | $\mathcal{N}(\mu, \sigma)$ |
| Blocked Walker | 18 | 6 | X-Coordinate $\geq$ -3 | $\mathcal{N}(\mu, \sigma)$ |
| Blocked Swimmer | 10 | 2 | X-Coordinate $\leq$ 0.5 | $\mathcal{N}(\mu, \sigma)$ |
| Biased Pendulum | 4 | 1 | X-Coordinate $\geq$ -0.015 | $\mathcal{N}(\mu, \sigma)$ |

balance. The game terminates when a maximum time step of 500 is reached. Similarly to the Blocked Half-Cheetah environment, we establish a constraint that blocks the region with X-coordinate $< -3$. As a result, the robot is only permitted to move within the region where the X-coordinate $\geq -3$.

3) *Blocked Walker.* In this environment, the agent controls a two-legged robot and learns how to make it walk. The termination conditions for the game are either when the robot loses its balance or when it reaches the maximum time step of 500. The reward obtained by the agent is determined by the distance the robot walks between two time steps, along with a penalty based on the magnitude of the input action. Similar to the above environments, we impose a constraint that blocks the region where the X-coordinate $< -3$. Consequently, the robot is only allowed to move within the region where the X-coordinate $\geq -3$.

4) *Blocked Swimmer.* In this environment, the agent controls a robot with two rotors connecting three segments, and learns how to move. The reward obtained by the agent is determined by the distance the robot walks between the current and previous time steps, along with a penalty based on the magnitude of the input action. The game ends when the robot reaches the maximum time step of 500. In contrast to the aforementioned environments, the Swimmer robot is easier to move forward than to move backward. Therefore, we constrain the region where the X-coordinate $> 0.5$. Consequently, the robot is only allowed to move within the region where the X-coordinate $\leq 0.5$.

5) *Biased Pendulum.* In this environment, the agent controls a pole to balance it on a cart. The game terminates either when the pole falls or when the maximum time step of 100 is reached. To increase difficulty, we assign higher rewards to the left locations, where the constraints are also imposed. Specifically, we introduce a constraint that blocks the region with X-coordinate $< -0.015$. At each step, a reward of 0.1 is provided if the X-coordinate $\geq 0$, and a reward of 1 is given if the X-coordinate $\leq -0.01$. The reward gradually decreases from 1 to 0.1 when the X-coordinate falls within $(-0.01, 0)$. Consequently, the agent is challenged to resist the temptation of higher rewards and stay within safe regions.

The MuJoCo environment settings in this work are summarized in Table C.1

### C.3 HIGHWAY DRIVING

Figure C.2 illustrates the Highway Driving environment. In this scenario, the ego car is displayed in blue while other cars are shown in red. The ego car has limited visibility and can only observe objects within its vicinity (marked in blue). The objective is to navigate the car to reach the destination point without going off-road, colliding with other cars, or violating time limits and other constraints (e.g., velocity).

In this paper, we mainly focus on the velocity constraint, where we limit the speed of the ego car to no more than 40 m/s, to ensure the ego car can drive at a safe speed. Additionally, we add Gaussian noise to the agent's action at each time step to introduce the control-level stochasticity.

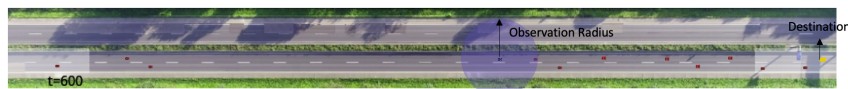

Figure C.2: The Highway Driving (HighD) environment.

# D   MORE EXPERIMENTAL RESULTS

## D.1   MORE RESULTS IN GRIDWORLD

Figure D.1 and Figure D.2 show the training curves in Gridworlds with the random rate $p_s = 0.01$ and $0.001$, respectively.

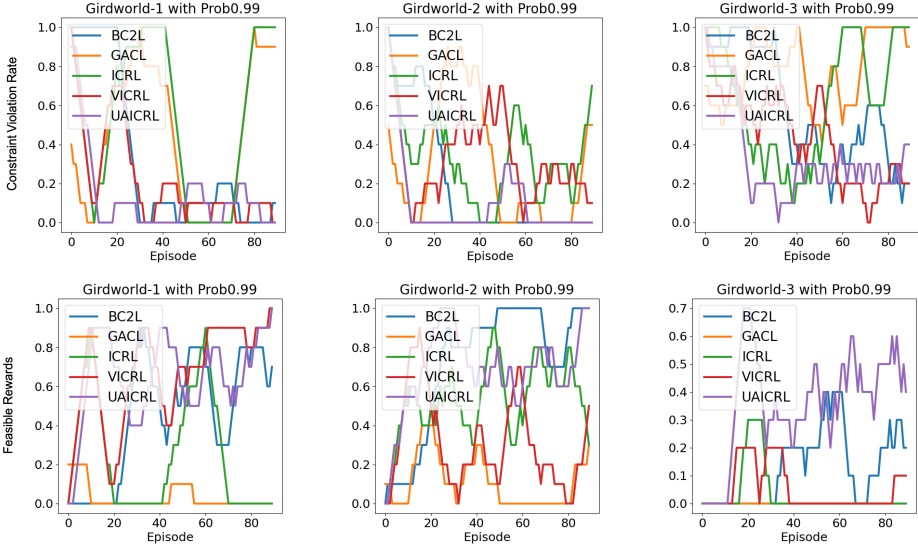

Figure D.1: The constraint violation rate (top) and feasible rewards (bottom) in three Gridworld settings with the random rate $p_s = 0.01$.

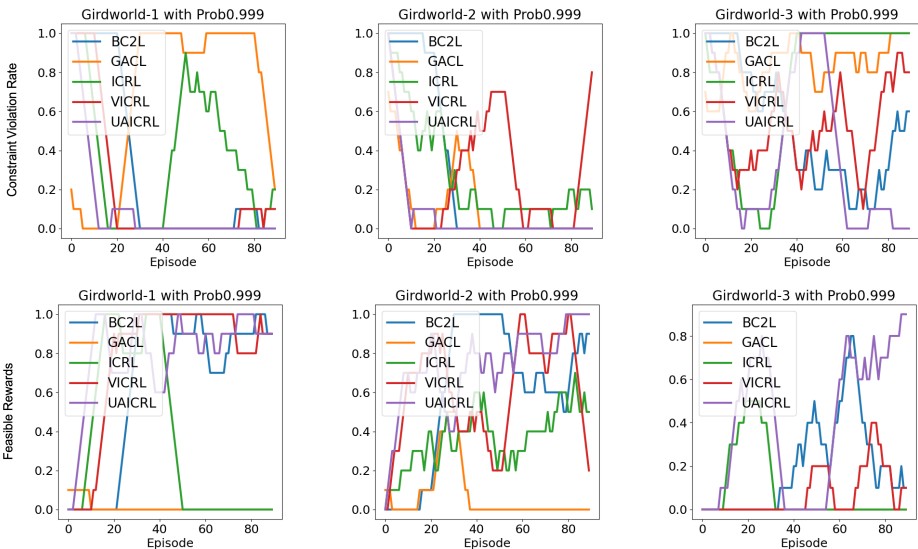

Figure D.2: The constraint violation rate (top) and feasible rewards (bottom) in three Gridworld settings with the random rate $p_s = 0.001$.

Figure D.3 illustrates the trajectories generated by PPO-Lag and DLPO in the last two Gridworld settings, along with the predicted constraint distributions.

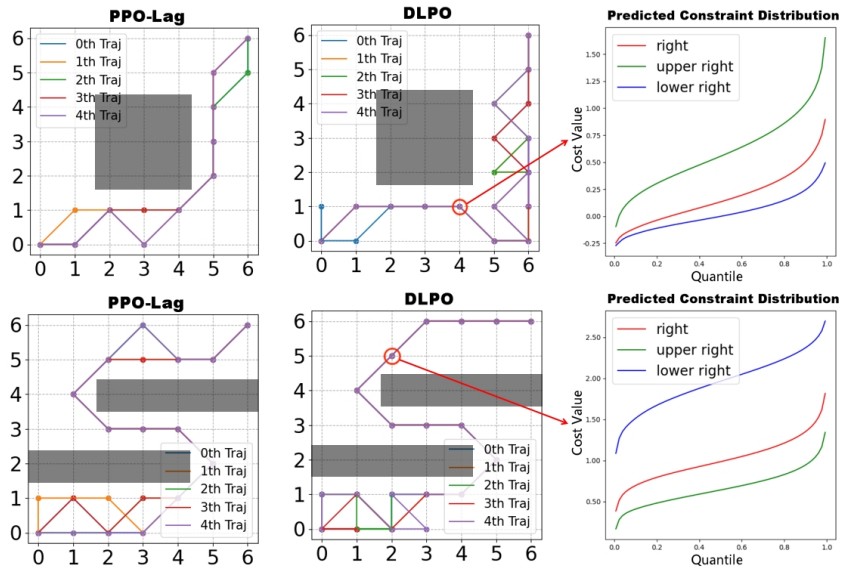

Figure D.3: Each row refers to a Gridworld scenario. We illustrate the trajectories generated by PPO-Lag and DLPO, along with the predicted cost distributions at the critical state denoted by red circles.

Figure D.4 illustrates the constraint map recovered by MEICRL and UAICRL in the last two Gridworld settings, along with the trajectories generated by the expert flow model. White circles denote the redundant constraints learned by MEICRL.

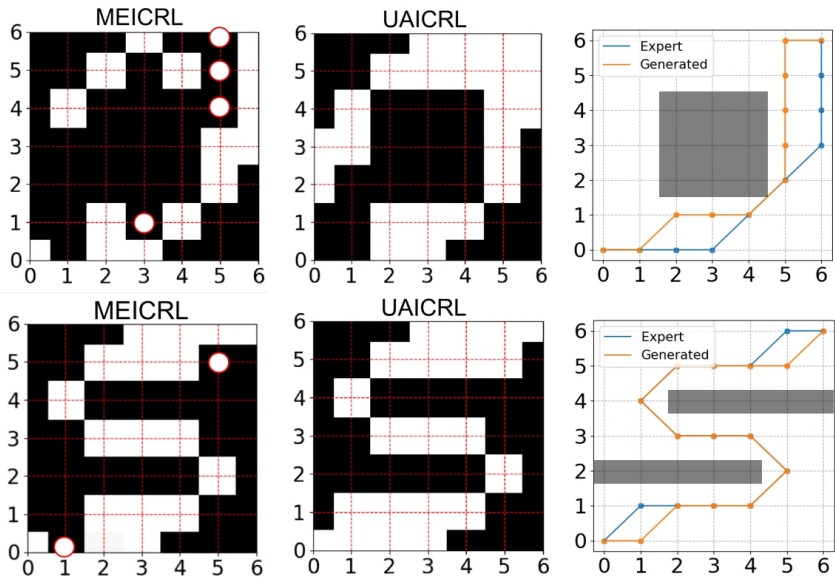

Figure D.4: Each row refers to a Gridworld scenario. We illustrate the constraint map recovered by MEICRL and UAICRL , along with the trajectories generated by the expert flow model.

## D.2    MORE RESULTS IN MUJOCO

Table D.1 shows the evaluation performance in MuJoCo environments with $\mathcal{N}(0, 0.1)$.

Table D.1: Evaluation of different methods in the MuJoCo environments with the stochasticity of $\mathcal{N}(0, 0.1)$. We report the mean ± std results averaged over 4 random seeds.

|  |  | Blocked Half-Cheetah | Blocked Ant | Blocked Walker | Blocked Swimmer | Biased Pendulum |
|---|---|---|---|---|---|---|
| Feasible Rewards | BC2L | 333.4±175.2 | 595.0±225.2 | 222.1±94.9 | 91.9±46.1 | 0.1617±0.1295 |
|  | GACL | 577.2±135.4 | 387.5±205.1 | 55.5±5.5 | 64.7±30.6 | 0.0572±0.0439 |
|  | ICRL | 343.0±224.2 | 965.0±432.4 | 166.9±49.0 | 91.0±42.8 | 0.1440±0.1184 |
|  | VICRL | 615.4±299.9 | 641.8±297.4 | 91.5±25.3 | 105.5±29.1 | 0.1447±0.1139 |
|  | UAICRL | **1714.4±222.0** | **1313.5±533.2** | **399.8±106.3** | **177.3±30.7** | **0.2551±0.1347** |
| Constraint Violation Rate | BC2L | 0.69±0.20 | 0.31±0.16 | 0.0±0.0 | 0.89±0.05 | 0.91±0.07 |
|  | GACL | 0.29±0.16 | **0.29±0.14** | 0.0±0.0 | 0.92±0.04 | 0.87±0.10 |
|  | ICRL | 0.67±0.21 | 0.52±0.18 | 0.0±0.0 | 0.90±0.05 | 0.93±0.05 |
|  | VICRL | 0.57±0.21 | 0.37±0.17 | 0.0±0.0 | 0.88±0.03 | 0.91±0.07 |
|  | UAICRL | **0.07±0.06** | 0.42±0.17 | **0.0±0.0** | **0.86±0.03** | **0.52±0.20** |

Figure D.5 and Figure D.6 show the additional experimental results in MuJoCo environments with constraint recovery under the stochasticity of $\mathcal{N}(0, 0.01)$ and $\mathcal{N}(0, 0.001)$, respectively.

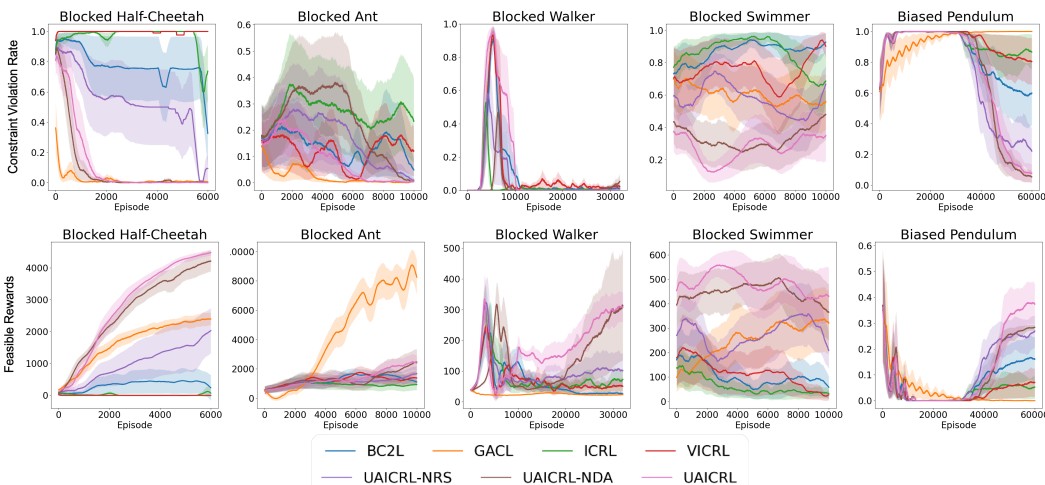

Figure D.5: The constraint violation rate (top) and feasible rewards (bottom) in MuJoCo environments with constraint recovery under the stochasticity of $\mathcal{N}(0, 0.01)$.

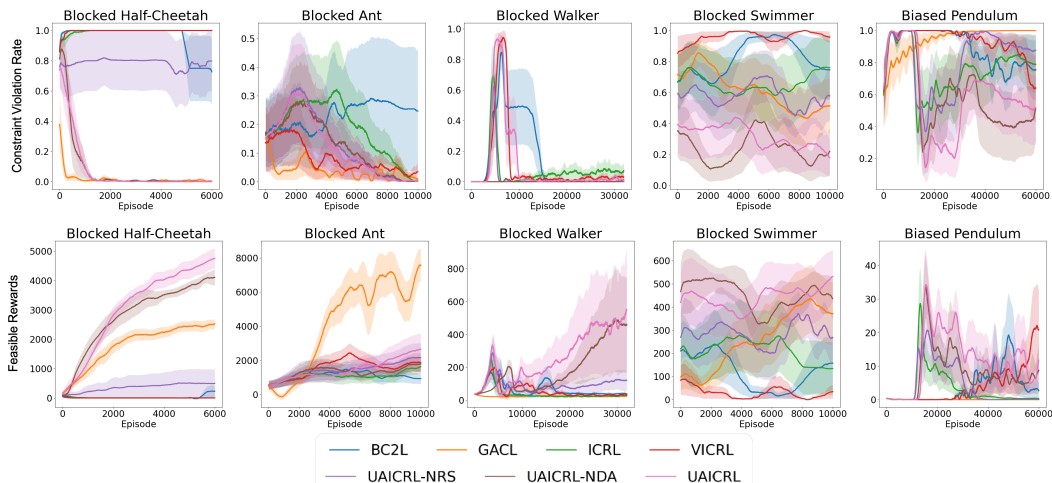

Figure D.6: The constraint violation rate (top) and feasible rewards (bottom) in MuJoCo environments with constraint recovery under the stochasticity of $\mathcal{N}(0, 0.001)$.

Figure D.7, D.8, and D.9 show the complete results in MuJoCo environments when given ground-truth constraints under the stochasticity of $\mathcal{N}(0, 0.1)$, $\mathcal{N}(0, 0.01)$ and $\mathcal{N}(0, 0.001)$, respectively.

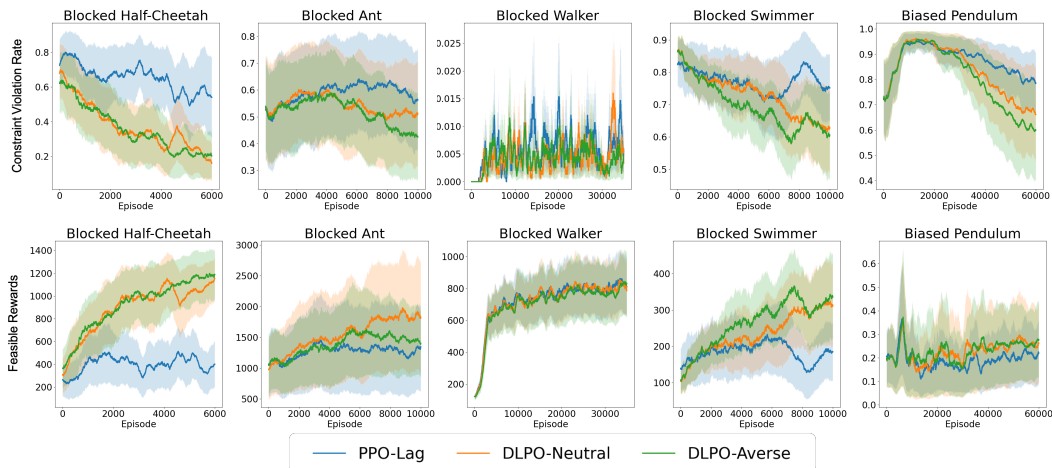

Figure D.7: The constraint violation rate (top) and feasible rewards (bottom) in MuJoCo environments with the stochasticity of $\mathcal{N}(0, 0.1)$ when given ground-truth constraints.

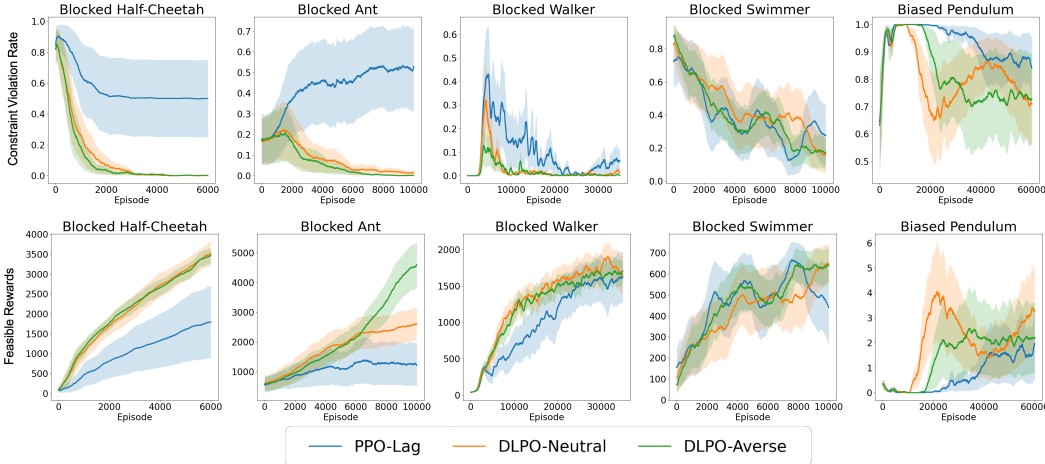

Figure D.8: The constraint violation rate (top) and feasible rewards (bottom) in MuJoCo environments with the stochasticity of $\mathcal{N}(0, 0.01)$ when given ground-truth constraints.

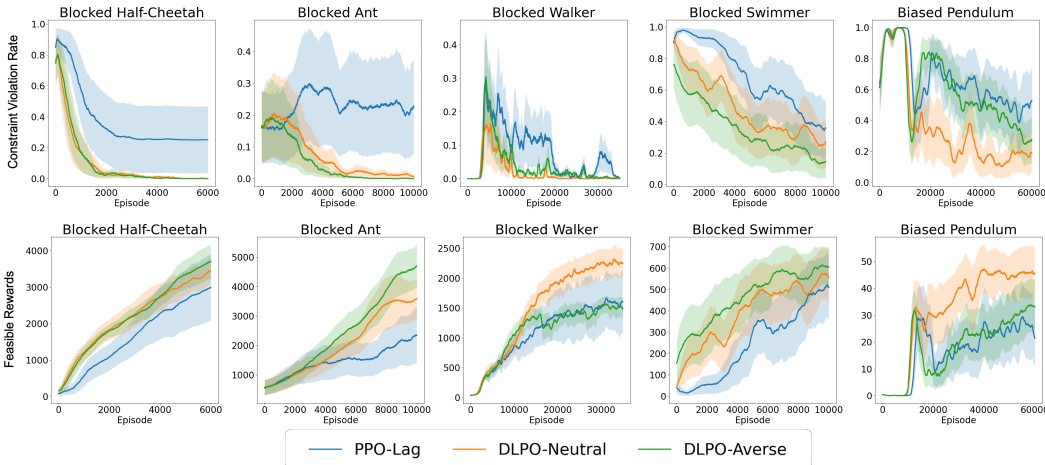

Figure D.9: The constraint violation rate (top) and feasible rewards (bottom) in MuJoCo environments with the stochasticity of $\mathcal{N}(0, 0.001)$ when given ground-truth constraints.

### D.3    MORE RESULTS IN HIGHD

Figure D.10 shows the additional experimental results in the HighD environment with constraint recovery under the stochasticity of $\mathcal{N}(0, 0.01)$ and $\mathcal{N}(0, 0.001)$, respectively.

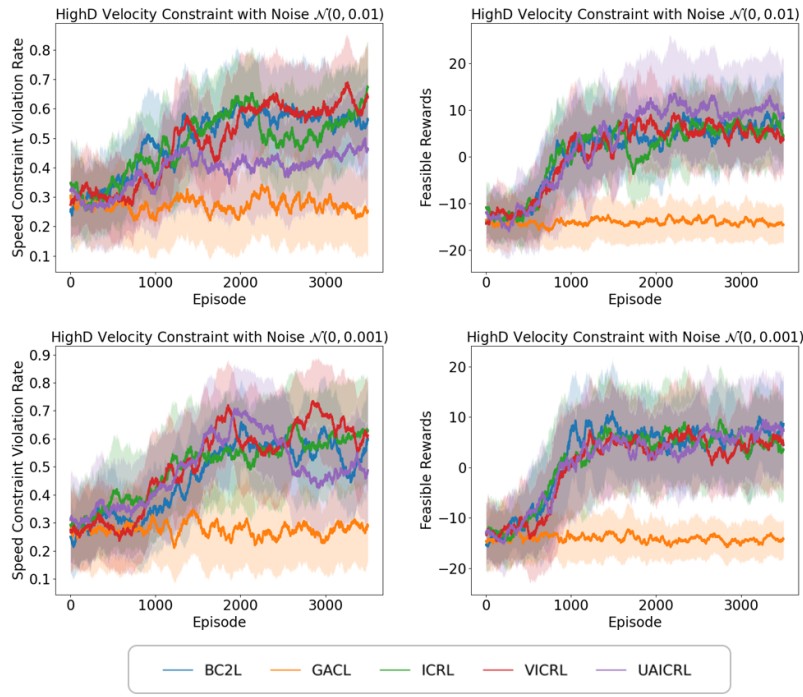

Figure D.10: Model performance in the HighD environment with velocity constraint under $\mathcal{N}(0, 0.01)$ (top) and $\mathcal{N}(0, 0.001)$ (bottom).

### D.4    MULTIPLE CONSTRAINT RECOVERY

To investigate the model performance under multiple constraint scenarios, we expand the HighD environment to incorporate both speed and distance constraints by generating an expert dataset, wherein the agent adheres to both constraints. Figure D.11 shows the results under the stochasticity of $\mathcal{N}(0, 0.1)$. We find that UAICRL still outperforms other baselines with the highest rewards while keeping a low constraint violation rate of both constraints, demonstrating its ability of uncertainty awareness with respect to multiple constraints. In addition, in comparison with the single constraint scenario (as shown in Figure 4), the rewards accrued by the policies exhibit a significant reduction, while the rate of constraint violations remains unaffected.

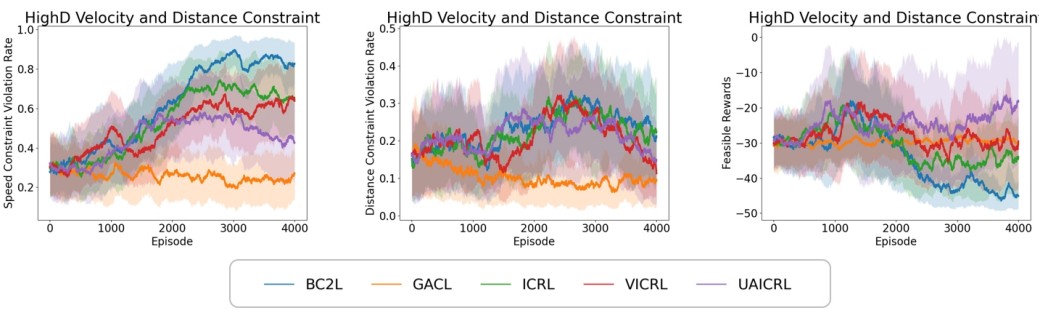

Figure D.11: Model performance in the HighD environment with both the speed and distance constraints under $\mathcal{N}(0, 0.1)$.

# E    MORE COMPREHENSIVE ANALYSIS

In this section, we provide a comprehensive analysis of the experimental results to supplement the discussion presented in the main body of the paper (Section 6.2). The aim is to delve deeper into the nuances of our experiments, offering a more detailed examination of the insights that could not be fully explored within the constraints of the main text. We summarize some meaningful findings and provide some explanations as follows:

*(1) "The baseline methods exhibit relatively unsatisfactory results in the experiments."*

This is because this paper primarily investigates the robustness of various methods in the face of aleatoric and epistemic uncertainties. As such, we intentionally control the scale of noise and the number of demonstrations, as detailed in our experimental settings (Section C). The baseline methods partially or completely disregard these factors (refer to Table 1). Specifically, 1) baseline methods that do not address stochastic transition dynamics fail to learn constraints that ensure the safety of the learned policy. This oversight can negatively influence control performance. 2) If epistemic uncertainty in Out-of-Distribution (OoD) data points is not taken into account, it could lead to inaccurate cost predictions, subsequently affecting policy updates and overall performance.

*(2) "When comparing the results across different levels of stochasticity, larger noise does not necessarily result in poorer model performance."*

The reason is that when noise levels are relatively low, its impact becomes challenging for the distributional value function to detect, resulting in a constraint that is less sensitive to the risk. Conversely, as the level of noise (i.e., stochasticity) increases, the variance of the estimated distribution becomes prominent, thereby making the constraint more responsive to the risk of unsafe behaviors. Our algorithms can thus induce stricter constraint functions and thus a lower constraint violation rate.

*(3) "The divergence trend observed in the UAICRL method, particularly in the Blocked Walker task."*

As shown in the results, we find that different algorithms display varying trends in their reward and constraint violation rates throughout the training curves. This divergence is most noticeable in the Blocked Walker environment. For instance, by the end of training, the rewards for UAICRL and UAICRL-NDA progressively increase, while those for other methods tend to remain unchanged or even decrease. To understand this phenomenon, it's important to note that both UAICRL and UAICRL-NDA implement risk-sensitive constraints. These constraints enable the agent to more effectively manage the aleatoric uncertainty resulting from noise, thereby facilitating the learning of risk-aware policies. This could result in considerable improvement in policy updates, particularly when the agent encounters critical situations or locations. For instance, these constraints enable the agent to initially learn to maintain a safe distance from hazardous zones. This strategy allows the agent to better understand how to accrue rewards without entering dangerous events.

*(4) "Most methods struggle to learn a safe policy with low constraint violation rate in the Blocked Swimmer task."*

There are several factors contributing to the complexity of the Blocked Swimmer task: 1) The Swimmer environment itself is inherently challenging to solve by RL methods. Previous works report similar difficulties in RL control (Franceschetti et al., 2022). 2) The mechanical dynamics of the Swimmer Robot make it easier for the robot to move forward than backward. The block region is also defined in front of the robot (where the X-coordinate $> 0.5$). The intrinsic robust dynamics may strongly compel the robot to proceed forward, leading to constraint violations even in deterministic settings (refer to Figure 2 in Liu et al. (2023)). 3) Thirdly, the stochastic transition dynamics further complicate the safe control, even when agents are aware of the ground-truth constraints (refer to PPO-Lag in Figure D.7). The task becomes even more difficult when the agent is required to predict constraints from provided demonstrations. In conclusion, the Blocked Swimmer environment already presents a challenge for traditional constrained RL. The challenge is amplified when constraints need to be inferred (as in the ICRL setting) and when transition dynamics are stochastic.

*(5) "The discrepancy in the performance of GACL, which performs well in the Block Ant task but not in other tasks in Fig. D.5 and D.6."*

The observed discrepancy in GACL's performance, where it works well in the Blocked Ant task but underperforms in other environments, can be attributed to its approach to constraint learning. Unlike

traditional constrained RL problems, GACL directly appends the discriminator $logD(s, a)$ to the reward function as a penalty, resulting in a modified reward function $r'(s, a) = r(s, a) + logD(s, a)$. This modification allows us to investigate the impact of simple reward shaping on constraint learning. We find GACL performs well in the Blocked Ant since this environment has the highest dimension in all environments. In other environments with lower dimensions, the discriminator $logD(s, a)$ is prone to overfitting the imperfect data early in training. The log-probability penalty from this low-quality discriminator applies a substantial penalty to many feasible regions, rendering the algorithm incapable of learning. However, in high-dimensional environments (like the Ant task), the discriminator is less likely to overfit. As a result, the impact of early mistakes in the discriminator is less significant, enabling continuous learning.

*(6) "In some environments in Figure 3, UAICRL performs similarly to UAICRL-NDA."*

We observe that in environments with high-dimensional state-action space (HalfCheetah, Ant, and Walker), UAICRL significantly outperforms UAICRL-NDA with higher rewards and lower constraint violation rates. However, in environments with relatively lower state-action dimensions (Swimmer and Pendulum), the difference of performance between UAICRL and UAICRL-NDA is less significant. This is because, in the high-dimensional environment, a training dataset of a fixed size can only encompass a limited number of data points within the input space. In contrast, in simpler or lower-dimensional environments, the input space is smaller. As a result, a dataset of the same size can cover a larger proportion of data points. This increased coverage reduces the chances of Out-of-Distribution (OoD) data points and the impact of epistemic uncertainty. Consequently, the incorporation of the data augmentation component does not lead to a substantial enhancement in performance.

