# OpenReview forum: "Uncertainty-aware Constraint Inference in Inverse Constrained Reinforcement Learning"
_ICLR.cc/2024/Conference — ICLR 2024 poster_

### Official Review · Reviewer_zJng · 2023-10-27

**Soundness:** 3 good
**Presentation:** 1 poor
**Contribution:** 3 good
**Rating:** 6
**Confidence:** 3

**Summary:**

This paper is interested in Inverse Constrained Reinforcement Learning (ICRL), that is, simultaneously learning imitation policies whilst learning and adhering to the constraints respected by the expert.
The authors propose to incorporate the uncertainties arising from stochastic environments (aleatoric uncertainty), along with
the epistemic uncertainties arising from learning from limited data.
To this end, they propose to learn the cost model with distributional Bellman updates.
They then propose a flow-based generative data augmentation scheme to mitigate issues arising from epistemic uncertainty.
That is, the augmented trajectories should remain in regions of the learned model which can be predicted confidently (low epistemic uncertainty).
The method is tested in i) a discrete-action grid world environment and ii) five MuJoCo environments with additional noise.

**Strengths:**

Although I am not well-read in Inverse Constrained Reinforcement Learning (ICRL), this paper appears to have highlighted
an important problem: imitation policies should satisfy the learned constraints subject to both the uncertainty in the environment (aleatoric uncertainty) and the uncertainty arising from learning the constraint from limited data (epistemic uncertainty).
Capturing the aleatoric uncertainty with distributional Bellman updates seems like a good idea and makes sense intuitively.
Whilst I found the flow-based trajectory generation section hard to follow, I get the general idea and this seems like a sensible way to consider the epistemic uncertainty.
I also liked that throughout the paper the authors provide intuition for the math with real-world examples. This was nice.

**Weaknesses:**

The paper's biggest weakness is its presentation and clarity.
I was generally happy with the paper up until Section 6 (Empirical Evaluation).
I found this section particularly hard to follow.
I'm not entirely sure what the main points are that the authors are trying to show in this section.
I would suggest the authors try to summarize the main questions and introduce them at the start of section 6.
This gives the reader an idea of what to expect in the section, which makes for an easier read.

**Experiments**
Remember, readers are stupid, you should hold their hand and walk them through your figures.
For example, what does robust and superior performance allude to in this sentence:
"When implementing both techniques, UAICRL demonstrates even more robust and superior performance compared to other methods."
This could be made a lot easier for the reader with something like:
"When implementing both techniques, UAICRL (pink) generally obtains high feasible rewards (top row Fig. 3) whilst having a low constraint violation rate (bottom row Fig. 3). This demonstrates that UAICRL is more robust and has superior performance compared to other methods."

**Illegible figures**
Most figures are illegible due to being too small and the font size being too low.
This needs to be fixed before publication.


**Conclusion is very short..**
The conclusion is very short and feels rushed. Surely the authors have more to say here??

**Bolding**
What does the bolding in the tables show? Does it show statistical significance from a statistical test or something else? This should be clarified somewhere in the text.

**Code**
There is no README file in the code supplement so it is not clear how to setup the environment or how to run the experiments.
It would be good to at least have a notebook to see how the code/method works in practice.

In my opinion, the paper highlights an important problem, has a good technical contribution and has results which support the claims.
However, I do not think the paper can be published until:
- The experiments section is made clearer
- The figures are made legible
- The conclusion is written properly

**Minor corrections**:
- The paper has many textual citations in parentheses. For example "(Liu et al., 2023; Papadimitriou et al., 2023)" in paragraph 3. You should use \citet instead of \citep to remove the parentheses.
- In Section 2, what is $\mathcal{M}^{c_{\omega}}$? It's not defined anywhere.
- Figure 1.
  - The text is way too small.
  - It's also not clear where to start reading from. I think you should start reading from $\mathcal{D}_{e}$ so perhaps this should be mentioned in the caption.
- Page 3 footnote is missing a full stop.
- Section 4.2
  - $\mathcal{T}$ is never formally introduced.
  - $F(\cdot)$ is overloaded and confusing as it is shown as $F(\tau)$, $F(s_{t})$ and $F(s,a)$. I think you should distinguish the state flow $F(s_{t})$ from the trajectory flow function.
- Algorithm 1:
  - Does "sample nominal trajectories" imply interacting with the environment? If so, I would explicitly state this.
  - Is it right that the augmented data is discarded at each iteration?
- Table 1 text is way too small
- Figure 2 needs larger text
- Figures 3/4/5/D.3/D.4/D.5/D.6
  - Larger text
  - It only needs one legend.
  - The x-axis ticks are too close
  - The plot titles don't all need to say "with noise $\mathcal{N}(0,0.1)$"
- Figure 6 is not clear.
  - Each column refers to a grid world scenario so this should be on the figure and/or mentioned in the caption.
  - The text is way too small
  - What are the constraints locations???
  - Where does the agent start/end?
- Figure 7 is not clear.
  - Each column refers to a grid world scenario so this should be on the figure and/or mentioned in the caption.
  - What does each row represent? Is the top row ICRL and the bottom row UAICRL? This needs to be made clearer.

**Questions:**

- What are the main questions your results section is trying to answer? Can you summarize them in a few bullet points?
- Have you made the figures legible?

---

> ### Author Response · Authors · 2023-11-17
> **Author response to Reviewer zJng**
>
> Dear Reviewer, we sincerely value your time and effort in evaluating our work. We are grateful for your constructive feedback and are committed to making the necessary revisions to meet the publication standards. We have prepared comprehensive responses and clarifications to address each point you raised. We hope these responses can resolve your concerns.
>
> 1. '*I would suggest the authors try to summarize the main questions and introduce them at the start of section 6. This gives the reader an idea of what to expect in the section.*'
>
> **Response.** Thank you for your valuable suggestions. We have revised the paper by "summarizing the main questions and introducing them" at the start of Section 6 in the revised paper.
>
> 2. '*Experiments. You should hold their hand and walk them through your figures. For example ...*'
>
> **Response.** Thank you for your suggestion. We have revised our paper by directly referencing the results, thereby eliminating the need for readers to search for them. This enhances the readability and accessibility of our paper.
>
> 3. '*Illegible figures. Most figures are illegible due to being too small and the font size being too low. This needs to be fixed before publication.*'
>
> **Response.** Thanks for the comments. To tackle the figure legibility, we have changed the unclear figures to make them more illegible, including the font size, legend, title, and caption, etc.
>
> 4. '*Conclusion is very short. The conclusion is very short and feels rushed.*'
>
> **Response.** Thanks for mentioning this concern. Due to the page limit, we were not able to expand our discussion. In the revised version, with extra space, we have revised the conclusion section, where we have summarized the key findings, underlined the implications of our research, and clearly articulated future research directions.
>
> 5. '*Bolding. What does the bolding in the tables show? This should be clarified somewhere in the text.*'
>
> **Response.** Thanks for mentioning this concern. In this paper, Bolding indicates the results that are most optimal among the compared results.
>
> 6. '*Code. There is no README file in the code supplement so it is not clear how to setup the environment or how to run the experiments.*'
>
> **Response.** In fact, we view code publication as an important component of our work, and our preferred platform is Github. We've already set up a Github repository including a README.md for setting up the Python virtual environment and a requirement.txt for installing Python packages, which will be made public once the anonymous review period ends.
>
> 7. '*Minor corrections.*'
>
> **Response.** Thank you for your detailed review of this work. We have updated the paper according to your suggestions one by one as follows:
> - "*textual citations in parentheses*": we have updated the paper with true citation format.
> - "*$\mathcal{M}^{c_{\omega}}$ in Section 2*": It denotes the CMDP with constraint function $c_\omega$.
> - "*Figure 1*": We have made the text larger and added descriptions in the caption.
> - "*Page 3 footnote*": We have added the ".".
> - "*Notation in Section 4.2*": $\mathcal{T}$ denotes the trajectory set, and following [1], we slightly abuse $F$ to denote the "flow".
>  And actually, the trajectory flow is the integral of state flow.
> - "*Algorithm 1*": We have explicitly stated that "interacting with the environment". During implementation, we generate new data in each iteration, and the old augmented data is discarded.
> - "*Table 1, Figure 2's text*": We have increased the font size.
> - "*Figures 3/4/5/D.3/D.4/D.5/D.6*": We modified the figures with larger text, one legend, wider x-axis ticks, and proper titles to make the m more legible.
> - "*Figure 6 is not clear*": We have changed Figure 6 with clearer explanations, larger text, and added the constraint locations.
> - "*Figure 7 is not clear*": We have changed Figure 7 with clearer explanations and illustrations.
>
> 8. '*Question 1: What are the main questions your results section is trying to answer? Can you summarize them in a few bullet points?*'
>
> **Response.** We empirically evaluate the effectiveness of UAICRL by answering the following questions:
> - How well does UAICRL perform in discrete (Section 6.1) and continuous (Section 6.2) environments under varying degrees of stochasticity?
> - Can UAICRL handle more complex, real-world highway driving scenarios (Section 6.3)?
> - How well do the key components of UAICRL (DLPO and FTG) address the aleatoric and epistemic uncertainty, respectively (Section 6.4)?
>
> 9. '*Question 2: Have you made the figures legible?*'
>
> **Response.** With your suggestions and our self-motivation to make the paper more readable and better, we have made our utmost effort to make the figures legible.
>
> **References**
>
> [1] Li, Yinchuan, et al. "CFlowNets: Continuous Control with Generative Flow Networks." ICLR (2023).

---

> > ### Comment · Reviewer_zJng · 2023-11-18
> >
> > Thank you for updating the manuscript and for your detailed response. I will increase my score to a 6.

---

> > > ### Author Response · Authors · 2023-11-18
> > >
> > > We sincerely appreciate your constructive feedback and are grateful for the time and effort you've invested in reviewing our work. Your insights have been instrumental in helping us enhance the quality and clarity of our paper. Thank you very much!

---

### Official Review · Reviewer_nMsK · 2023-10-27

**Soundness:** 3 good
**Presentation:** 3 good
**Contribution:** 3 good
**Rating:** 6
**Confidence:** 3

**Summary:**

This paper presents a thorough discussion on the topic of inverse safe reinforcement learning. The authors introduce the Uncertainty-Aware Constraint Inference Constrained Reinforcement Learning (UAICRL), a novel framework that takes into account both aleatoric and epistemic uncertainties to enhance uncertainty awareness in constraint inference. The authors conducted extensive experiments to demonstrate the superior performance of their method over several other ICRL methods in both continuous and discrete environments, highlighting its strong uncertainty-aware capabilities.

**Strengths:**

(1) Noteworthy topic: The study on Inverse Constrained Reinforcement Learning appears to address a critical issue, and the proposed method holds promising potential for real-world applications.

(2) Extensive experimental validation: The authors' extensive experiments with a wide range of baseline methods and tasks to demonstrate the strength of their approach are commendable and impressive.

**Weaknesses:**

(1) Insufficient theoretical support: It is observed that the proposed method may benefit from further strengthening its theoretical foundations, as acknowledged by the authors.

(2) Limited discussion and explanation of experiments: While the manuscript presents extensive experimental results, a more comprehensive discussion and elaboration of these findings would enhance the paper's overall quality. Moreover, a detailed examination of performance across various scales of randomness within the primary context could provide valuable insights, as noted in my question (1-3).

**Questions:**

(1) Could you please explain the reason behind the divergence trend observed in the UAICRL method in section 6.2, particularly in the Blocked Walker task? Additionally, would it be possible to provide results with an extended number of epochs? Were the curves smoothed in your analysis?

(2) What factors contribute to the challenges posed by the Blocked Swimmer task? It seems that most methods struggle to learn a safe (low-cost-violation) policy for this specific task.

(3) Could you elaborate on the factors leading to the relatively unsatisfactory results of the baseline methods in the tasks? Specifically, what could explain the discrepancy in the performance of GACL, which performs well in the Block Ant task but not in other tasks shown in Figures D.3 and D.4?

(4) How would you describe the generalizability of your method to multi-cost settings?

---

> ### Author Response · Authors · 2023-11-17
> **Author response to Reviewer nMsK - Part 1**
>
> Dear Reviewer, we greatly appreciate your constructive comments. We have seriously considered your suggestions, and hopefully, the following response can address your concerns:
>
> 1. '*Limited discussion and explanation of experiments: While the manuscript presents extensive experimental results, a more comprehensive discussion and elaboration of these findings would enhance the paper's overall quality. Moreover, a detailed examination of performance across various scales of randomness within the primary context could provide valuable insights, as noted in my question (1-3).*'
>
> **Response.** We appreciate your insightful comments on the discussion and interpretation of our experimental results. We totally agree that a more exhaustive analysis would significantly enhance the quality of our paper.
>
> However, due to constraints on page limits, we were unable to provide detailed explanations for all observed phenomena. As a result, we chose to focus on some of the most intriguing and significant findings for further elaboration.
>
> In the revised version of our paper (see our following response and please refer to section 6 and Appendix E in our paper), we have conducted a more in-depth analysis of our results. We hope this could address your concerns.
>
> 2. '*Could you please explain the reason behind the divergence trend observed in the UAICRL method in section 6.2, particularly in the Blocked Walker task?*'
>
> **Response.**
> Thank you for bringing up this concern. By "divergence trend," we assume you're referring to the observed pattern where different baseline algorithms display varying trends in their reward and constraint violation rates throughout the training curves. This divergence is most noticeable in the Blocked Walker environment. For instance, by the end of training, the rewards for UAICRL and UAICRL-NDA progressively increase, while those for other methods tend to remain unchanged or even decrease.
>
> To understand this phenomenon, it's important to note that **both UAICRL and UAICRL-NDA implement risk-sensitive constraints**. These constraints enable the agent to more effectively manage the aleatoric uncertainty resulting from noise, thereby facilitating the learning of risk-aware policies. This could result in considerable improvement in policy updates, particularly when the agent encounters critical situations or locations.
> For instance, these constraints enable the agent to initially learn to maintain a safe distance from hazardous zones. This strategy allows the agent to better understand how to accrue rewards without entering dangerous events.
>
> 3. '*Additionally, would it be possible to provide results with an extended number of epochs? Were the curves smoothed in your analysis?*'
>
> **Response.** To delve deeper into this phenomenon, as suggested by the reviewer, we have extended the number of epochs in the Blocked Walker environment, subjecting it to the noise of $\mathcal{N}(0,0.1)$ (see Blocked Walker in Figure 3). In this specific task, we observed that the trend lines become smooth when the episode count reaches 60k. Furthermore, the rewards for both UAICRL and UAICRL-NDA begin to increase at a higher rate compared to the other methods. We have revised the paper by adding these updates.
>
> 4. '*What factors contribute to the challenges posed by the Blocked Swimmer task? It seems that most methods struggle to learn a safe (low-cost-violation) policy for this specific task.*'
>
> **Response.**  Thank you for bringing up this concern. There are, in fact, several factors contributing to the complexity of the Blocked Swimmer task:
>
> - Firstly, the Swimmer environment itself is inherently challenging to solve by RL methods. Previous works report similar difficulties in RL control [1].
>
> - Secondly, the mechanical dynamics of the Swimmer Robot make it easier for the robot to move forward than backward. The block region is also defined in front of the robot (where the X-coordinate $>0.5$). The intrinsic robust dynamics may strongly compel the robot to proceed forward, leading to constraint violations even in deterministic settings (refer to Figure 2 in [2]).
>
> - Thirdly, the stochastic transition dynamics further complicate the safe control, even when agents are aware of the ground-truth constraints (refer to PPO-Lag in Figure D.7). The task becomes even more difficult when the agent is required to predict constraints from provided demonstrations.
>
> In conclusion, the Blocked Swimmer environment already presents a challenge for traditional constrained RL. The challenge is amplified when constraints need to be inferred (as in the ICRL setting) and when transition dynamics are stochastic.
>
> **References**
>
> [1] Franceschetti, Maël, et al. "Making Reinforcement Learning Work on Swimmer." arXiv preprint arXiv:2208.07587 (2022).
>
> [2] Liu, Guiliang, et al. "Benchmarking constraint inference in inverse reinforcement learning." ICLR (2023).

---

> ### Author Response · Authors · 2023-11-17
> **Author response to Reviewer nMsK - Part 2**
>
> 5. '*Could you elaborate on the factors leading to the relatively unsatisfactory results of the baseline methods in the tasks?*'
>
> **Response.**
> Our paper primarily investigates the robustness of various methods in the face of aleatoric and epistemic uncertainties. As such, we intentionally control the scale of noise and the number of demonstrations, as detailed in our experimental settings. The baseline methods partially or completely disregard these factors (refer to Table 1). Specifically,
>
> - Baseline methods that do not address stochastic transition dynamics fail to learn constraints that ensure the safety of the learned policy. This oversight can negatively influence control performance.
>
> - If epistemic uncertainty in Out-of-Distribution (OoD) data points is not taken into account, it could lead to inaccurate cost predictions, subsequently affecting policy updates and overall performance.
>
> 6. '*Specifically, what could explain the discrepancy in the performance of GACL, which performs well in the Block Ant task but not in other tasks shown in Figures D.3 and D.4?*'
>
> **Response.** The observed discrepancy in GACL's performance, where it works well in the Blocked Ant task but underperforms in other environments, can be attributed to its approach to constraint learning. Unlike traditional constrained RL problems, GACL directly appends the discriminator $logD(s, a)$ to the reward function as a penalty, resulting in a modified reward function $r'(s, a) = r(s, a) + logD(s, a)$. This modification allows us to investigate the impact of simple reward shaping on constraint learning.
>
> We find GACL performs well in the Blocked Ant since this environment has the highest dimension in all environment. In other environments with lower dimensions, the discriminator $logD(s, a)$ is prone to overfitting the imperfect data early in training. The log-probability penalty from this low-quality discriminator applies a substantial penalty to many feasible regions, rendering the algorithm incapable of learning.
> However, in high-dimensional environments (like the Ant task), the discriminator is less likely to overfit. As a result, the impact of early mistakes in the discriminator is less significant, enabling continuous learning.
>
> 7. '*How would you describe the generalizability of your method to multi-cost settings?*'
>
> **Response.**
> Thanks for raising this concern.
> We assume the term "multi-cost settings" indicates the ground-truth CMDP has different kinds of constraints, and the question is "How well our neural constraint function can approximate these different kinds of constraints".
>
> We follow a recent paper [1] and study the performance of this neural constraint function in the autonomous driving environment under our UAICRL framework. The experiment results and the corresponding analyses are included in Appendix D.4. The results show that UAICRL outperforms other baselines with the highest rewards while keeping a low violation rate of both types of constraints, demonstrating its ability of uncertainty awareness with respect to multiple constraints.
>
> [1] Liu, Guiliang, et al. "Benchmarking constraint inference in inverse reinforcement learning." ICLR (2023).

---

> > ### Comment · Reviewer_nMsK · 2023-11-18
> > **Thanks for the response**
> >
> > Thank you for updating the manuscript and addressing my concerns! After reading reviews provided by other reviewers and the authors' responses, I will maintain my score.

---

> > > ### Author Response · Authors · 2023-11-20
> > >
> > > Thank you for evaluating our work and for your insightful feedback throughout this process. We appreciate the time you have dedicated to reviewing our work.

---

### Official Review · Reviewer_8mMN · 2023-10-29

**Soundness:** 3 good
**Presentation:** 3 good
**Contribution:** 2 fair
**Rating:** 6
**Confidence:** 3

**Summary:**

This paper considers addressing the uncertainty issues in the inverse constrained RL problem. The authors propose to (1) replace cost critic by a distributional one in constrained RL to model the aleatoric uncertainty, and (2) use FTG to augment data to reduce epistemic uncertainty. The authors compare their method with previous inverse constrained RL baselines on different domains including gridworld, safety-mujoco and highway driving.

**Strengths:**

- This paper is well organized and easy to follow.
- The proposed method addresses two types of uncertainties in constrained RL, which are overlooked by previous research.
- The ablation studies validate the effectiveness of distributional critic when encountering aleatoric uncertainty.

**Weaknesses:**

- The effectiveness of data augmentation is a little questionable.
    - In theory, FTG can augment the expert and nominal datasets but the last term in the objective of eq.(8) includes the OOD trajectories $\bar{\tau}$. So how do you generate $\bar{\tau}$?
    - In practical experiment (fig 3), UAICRL actually performs similarly to UAICRL-NDA, which removes the data augmentation part.
    - Although the authors give more illustrations in fig.7 (I suppose the top is for MEICRL and bottom is for UAICRL), I think it's not very clear. For example, I believe the authors should at least explain what the generated trajectory is, and which parts are OOD.

minor issues:
- In fig 4, the baseline should be "GACL" instead of "GAIL".

**Questions:**

- What are the target cost limits $\epsilon$ for experiments in table 2, fig 3&5?
- Why are some experiments early stopped when they obviously have not converged? E.g., in fig.3&4.
- The authors run experiments on Mujoco tasks with different scales of stochasticity in env. But many baselines have much higher constraint violation rate with smaller noise, e.g., comparing fig.3, D3&D4. My intuition is that these methods should behave better with smaller noise as they cannot model such uncertainty. Could you explain it?

---

> ### Author Response · Authors · 2023-11-17
> **Author response to Reviewer 8mMN - Part 1**
>
> We greatly appreciate your thought-provoking questions. It is our sincere hope that the answers provided will effectively address and alleviate your concerns.
>
> 1. '*In theory, FTG can augment the expert and nominal datasets but the last term in the objective of eq.(8) includes the OOD trajectories. So how do you generate?*'
>
> **Response.**
> Thanks for your comments. Our FTG can generate a rich number of trajectories that are not included in the original dataset (OOD) due to its ability of:
>
> - **Generating Diverse Samples.**
> Intuitively, FTG is based on Generative Flow Networks (GFlowNets), which demonstrated excellent performance in generating a diverse array of samples in the realm of scientific discovery ([1]). Rather than consistently producing the highest-scoring trajectories, which correspond to the largest flows, the generation process employed by GFlowNets is characterized by sampling from a distribution represented by learned flow functions. Many of the samples are **new**, and thus they can facilitate scientific discovery.
>
> - **Generating OoD Trajectories.**
> Conceptually, at each time step, the state-action pairs that are sampled reflect the patterns **observed in the provided dataset** (for example, molecules). However, the true strength of this approach is realized when **these pairs are concatenated**. This process enables the representation of **Out-of-Distribution (OoD) trajectories** (for example, proteins), extending the utility of the GFlowNets beyond the confines of the training data.
>
> - **Generating OoD Data in Feature Space.**
> Many previous works ([2][3]) demonstrate the effectiveness of generative models' (like GFlowNets, GAN, VAE, diffusion models) ability with out-of-distribution data. Additionally, it's crucial to note that the superior performance of generative models (like GPT) can be attributed to leveraging high-level latent representations of raw inputs. Similarly, we believe that our FTG exhibits comparable expressive power, generating trajectories that resemble the training data in the latent space, even if they are OoD in the state-action feature space.
>
> In conclusion, with efficient representing capabilities and objective-driven sampling, FTG is able to generate OoD data.
>
> 2. '*In practical experiment (fig 3), UAICRL actually performs similarly to UAICRL-NDA, which removes the data augmentation part.*'
>
> **Response.** Thanks for raising this concern. In fact, we observe that in environments with high-dimensional state-action space (HalfCheetah, Ant, and Walker), UAICRL significantly outperforms UAICRL-NDA with higher rewards and lower constraint violation rates. However, in environments with relatively lower state-action dimensions (Swimmer and Pendulum), the difference of performance between UAICRL and UAICRL-NDA is less significant.
> This is because, in the high-dimensional environment, a training dataset of a fixed size can only encompass a limited number of data points within the input space. In contrast, **in simpler or lower-dimensional environments**, the input space is smaller. As a result, a dataset of the same size can **cover a larger proportion of data points**. This increased coverage reduces the chances of Out-of-Distribution (OoD) data points and **the impact of epistemic uncertainty**. Consequently, the incorporation of a data augmentation component does not lead to a substantial enhancement in performance. We have ensured this observation is properly conveyed in the revised paper (see Q6 in Appendix E).
>
> 3. '*Although the authors give more illustrations in fig.7, I think it's not very clear. For example, I believe the authors should at least explain what the generated trajectory is, and which parts are OOD.*'
>
> **Response.** Thanks for the suggestions. We have added an example of generated trajectories in Figure 7 and Figure D.4 in the Appendix. In the paper, we have explained how the generated trajectory increases the diversity of the dataset, thus reducing the impact of epistemic uncertainty. The generated trajectory is not incorporated into the expert dataset due to the restricted size of dataset. This trajectory enhances the diversity of the dataset, thereby mitigating the redundant constraints learned by MEICRL.
>
> 4. '*What are the target cost limits $\epsilon$ for experiments in table 2, fig 3/5?*'
>
> **Response.** Thank you for your suggestion. In reality, we have set $\epsilon$ to 1e-8. This setting is significant in the context of ICRL for two reasons: 1) When $\epsilon=0$, it represents a hard constraint that necessitates absolute satisfaction. 2) Conversely, when $\epsilon>0$, it represents soft constraints that allow a degree of constraint violation. Considering our primary focus on stochastic environments, where achieving absolute constraint satisfaction can be challenging, we have chosen a small, yet positive value for $\epsilon$. This allows a manageable degree of constraint violation, aligning with the setting of our environments.

---

> > ### Author Response · Authors · 2023-11-17
> > **Author response to Reviewer 8mMN - Part 2**
> >
> > 5. '*Why are some experiments early stopped when they obviously have not converged? E.g., in fig.3/4.*'
> >
> > **Response.** Thank you for bringing up this issue. We interpret "convergence" to mean the point (i.e., episode) at which the performance of the algorithms stabilizes or remains unchanged with further training. Yet, in a stochastic environment, the training of the algorithm is consistently influenced by noise, which makes achieving convergence more challenging. In our experiment, defining an upper bound for the convergence episode across various random seeds (since the added noise is affected by these seeds) and tested algorithms is challenging.
> >
> > To address this issue, we empirically identify such an episode at which the algorithm's performance neither significantly decreases nor increases. The plots in Figures 3 and 4 were generated according to this criterion. Even though strict convergence may not have been achieved, these training plots are particularly valuable for illustrating sample complexity. An algorithm with lower sample complexity is expected to converge more rapidly.
> >
> > But to address your concerns, we conduct additional experiments with an extended number of epochs for some environments that seem not to converge (i,e, Half-cheetah, Pendulum, and Walker with noise $\mathcal{N}(0,0.1)$), and we have updated the plots in the revised paper (see Figure 3).
> >
> > 6. '*The authors run experiments on Mujoco tasks with different scales of stochasticity in env. But many baselines have much higher constraint violation rate with smaller noise, e.g., comparing fig.3, D3/D4. My intuition is that these methods should behave better with smaller noise as they cannot model such uncertainty. Could you explain it?*'
> >
> > **Response.** Thanks for raising this concern. In fact, we have briefly mentioned this finding in the last two lines in Section 6.2, we have updated the paper and discussed this finding in more detail (see Q2 in Appendix E). To elaborate, when noise levels are relatively low, its impact becomes challenging for the distributional value function to detect, resulting in a constraint that is less sensitive to the risk. Conversely, as the level of noise (i.e., stochasticity) increases, the **variance** of the estimated distribution becomes prominent, thereby making the constraint more responsive to the risk of unsafe behaviors. Our algorithms can thus induce stricter constraint functions and thus a lower constraint violation rate.
> >
> > 7. '*In fig 4, the baseline should be "GACL" instead of "GAIL".*'
> >
> > **Response.** Thanks for your correction! We have updated it in the revised paper.

---

> ### Comment · Reviewer_8mMN · 2023-11-19
>
> Thanks for your reply. I have some further questions as follows:
>
> Q4&5 You set $\epsilon=1e-8$ but it seems that the results on very few tasks achieve that cost constraint $E[\sum\gamma^t c_t]\leq \epsilon$. I guess that can also partially explain why you need to early stop the training. In my opinion, the cost limit you set is almost impossible for current method to achieve, not to mention the gap between the predicted and true costs in context of inverse RL. Therefore, I think probably you can set a large threshold instead, then the constrained policy optimization in eq.(4) is at least feasible.
>
>
> Q6 You seemed to misunderstand my question. I was talking about the performances of **baselines**, which does not have distributional value function. The performances of baselines with large noise seem to be better than those with small noise (e.g., Blocked Halfcheetah), which is not intuitive. So why does it happen?

---

> ### Author Response · Authors · 2023-11-20
> **Author response to Reviewer 8mMN - Part 1**
>
> Thank you for your valuable feedback. We greatly appreciate your important concerns and suggestions. We believe these concerns are addressable with the following points:
>
> 1. *'Q4/5. You set $\epsilon=1e-8$ but it seems that the results on very few tasks achieve that cost constraint $E[\sum\gamma^tc_t]\leq\epsilon$. I think probably you can set a large threshold instead, then the constrained policy optimization in eq.(4) is at least feasible.'*
>
> **Response.**
> Thanks for raising this concern. It appears determining the threshold is indeed a challenging problem. Our approach involves setting it to a small number ($\epsilon=1e-8$). In our experiment, we have conducted investigations into the expected discounted cumulative cost $E_{\pi^E}[\sum_t^T\gamma^t c_t]$ in expert policies under varying noise levels. We discovered that most of these values are very close to $1e-8$. We acknowledge that finding a policy satisfying $E_{\pi}[\sum_t^T\gamma^t c_t]\leq E_{\pi^E}[\sum_t^T\gamma^t c_t]$ doesn't necessarily imply an absolute adherence to constraints (as seen in many of our results) since the expert policy can not necessitate zero constraint violation rate in the stochastic environment. Nevertheless, we believe it can serve as a suitable reference for determining $\epsilon$.
>
> If we set $\epsilon$ to a larger number, which significantly impacts optimization, it introduces considerable complexity to the forward constrained optimization problem. For instance, to solve the forward problem with an algorithm like PPO that learns the policy with value functions, we need to distribute $\epsilon$ to all the state and time steps (i.e., be able to compute $\epsilon_t(s)$ for all states and time steps). To determine $\pi(a|s)$, the algorithm must ensure that $\rho_{\alpha}\Big[\sum_{\iota=t}^{T}\gamma^{\iota}C(S_\iota,a_\iota)|s\Big] \le \epsilon_t(s)$ at time step $t$ for state $s$. To circumvent this complexity, researchers often study the hard constraint ($\epsilon=0$) or set $\epsilon$ to a small number that allows $\epsilon_t(s)=\epsilon,~ \forall{s,t}$.
>
> 2. *'Q6 The performances of baselines with large noise seem to be better than those with small noise (e.g., Blocked Halfcheetah), which is not intuitive. So why does it happen?'*
>
> **Response.** Thanks for raising this issue, which drives us to spend time re-checking the rationality of our experiment settings and results. Now, we can ensure that we have maintained the consistency of hyperparameters among baseline methods, which ensured the correctness and reproducibility of these experiments (so that your observation can be reproduced). To answer your concerns, we additionally perform some studies in the Half-Cheetah environment, based on which we observe several phenomena. The mean results over 4 trials can be found in the following tables (please refer to Table 2 in the next response part for results under noise of $\mathcal{N}(0,0.001)$), where 'seed' denotes the seed of environmental noise (0 in this work), 'hidden size' denotes the hidden layer of the constraint network, and UAICRL-NDA shares the same hyperparameters with other baselines except for the distribution estimator.
>
> **Table 1. Model performance under noise of $\mathcal{N}(0,0.01)$.**
> |                            |            | seed=0, hidden size=20(original) | seed=0, hidden size=64 | seed=100, hidden size=20 |
> |:--------------------------:|:----------:|:--------------------------------:|:----------------------:|:------------------------:|
> |      **Feasable  Rewards**     | UAICRL-NDA |              4232.8              |         3967.8         |          3597.5          |
> |                            |    ICRL    |               99.2               |          167.9         |           131.9          |
> |                            |    BC2L    |               274.1              |          398.7         |           804.9          |
> |                            |    VICRL   |                 0.0                |          121.6         |           78.1           |
> | **Constraint  Violation Rate** | UAICRL-NDA |               0.21%              |          0.42%         |           0.60%          |
> |                            |    ICRL    |               70.3%              |          59.2%         |           56.1%          |
> |                            |    BC2L    |               40.7%              |          29.8%         |           27.8%          |
> |                            |    VICRL   |               100%               |          75.6%         |           89.7%          |

---

> ### Author Response · Authors · 2023-11-20
> **Author response to Reviewer 8mMN - Part 2**
>
> **Table 2. Model performance under noise of $\mathcal{N}(0,0.001)$.**
> |                            |            | seed=0, hidden size=20(original) | seed=0, hidden size=64 | seed=100, hidden size=20 |
> |:--------------------------:|:----------:|:--------------------------------:|:----------------------:|:------------------------:|
> |      **Feasable  Rewards**     | UAICRL-NDA |              4100.2              |         4338.2         |          3911.5          |
> |                            |    ICRL    |                0.0               |          212.1         |           40.2           |
> |                            |    BC2L    |               227.6              |          362.3         |           187.8          |
> |                            |    VICRL   |                0.0               |          175.8         |           118.6          |
> | **Constraint  Violation Rate** | UAICRL-NDA |               0.08%              |          0.09%         |           0.04%          |
> |                            |    ICRL    |               100%               |          65.7%         |           69.6%          |
> |                            |    BC2L    |               74.1%              |          32.3%         |           76.8%          |
> |                            |    VICRL   |               100%               |          71.9%         |           78.1%          |
>
> By analyzing the results, we provide some potential reasons:
>
> - Different Level of Exploration: Note that our work follows the commonly applied Gaussian representation for policy in continuous action space. In a high-noise environment, the policy model appears to be more inclined towards exploration, as it's less certain about the outcomes and the corresponding variance term in the Gaussian becomes larger. This can lead to a more thorough understanding of the dynamics in the environment. In low-noise environments, the model might exploit known strategies more, potentially missing out on understanding the full breadth of constraints.
>
> - Larger noise prevents overfitting in early training: In supervised learning, a common strategy to prevent overfitting involves injecting random noise into the input samples. Similarly, we anticipate that the RL policy will need to adapt to the noise in environmental dynamics during reward maximization. This adaptation could result in improved generalization to unpredictable future noise. In environments with lower noise levels, the model converges fast, but to a policy that may not be as effective. In this work, we initially followed the hyperparameters from the benchmark which studied deterministic environments, and set the architecture of the constraint net with 20 hidden dimensions across the Half-Cheetah environments. As shown in the table, when the hidden size of the constraint net is larger, it alleviates the overfitness and performs relatively better than the small network, although the improvement is not very significant.
>
> - The random seeds in the environment can affect the initial conditions and subsequent evolution of the learning process. This can include variations in the initial exploration of the environment, the first few updates to the policy, and the initial understanding of constraints.
>
> In summary, we find that almost all baseline methods struggle to achieve satisfactory performance in the Half-Cheetah environment with small noise. This further demonstrates the significance of handling the noise in the ICRL framework and the robustness of the proposed method. We have ensured this suboptimal performance of ICRL under smaller noise can happen, and this reason can be studied in future work.

---

### Official Review · Reviewer_4NEv · 2023-10-31

**Soundness:** 3 good
**Presentation:** 3 good
**Contribution:** 3 good
**Rating:** 5
**Confidence:** 3

**Summary:**

This paper presents UAICRL, a novel approach for addressing Inverse Constrained Reinforcement Learning (ICRL) by considering both aleatoric and epistemic uncertainties. UAICRL leverages a distributional critic in conjunction with a risk-measure to calculate the cost, so as to handle aleatoric uncertainty. In addition, it utilizes mutual information and flow-based trajectory generation techniques to reduce epistemic uncertainty. The experimental results demonstrated improved performance and included ablation studies on the use of risk-sensitive constraint and data augmentation.

**Strengths:**

- Addresses both aleatoric and epistemic uncertainty, in contrast to previous methods that primarily focus on epistemic uncertainty.
- Works with both continuous and discrete spaces, unlike most previous methods limited to discrete spaces.
- Supports stochastic training environments, whereas earlier works frequently assume deterministic environments.

**Weaknesses:**

1. The ablation of the mutual information term in Eq. (7) results in a configuration where only the risk-sensitive constraint from Eq. (4) is utilized. This particular setup is not discussed in the paper.
2. I'm concerned about expanding the dataset by generating trajectories based on a learned flow function. It is still possible for the flow function to generate out-of-distribution data.

**Questions:**

1. I'm wondering if the flow functions can be substituted with other conditional generative models, or if the flow matching objective is tightly coupled with UAICRL. For example, can one use a conditional diffusion model to replace the Flow-based Trajectory Generation (FTG) algorithm?
2. While Table B.1 suggests that FTG can maintain consistent hyperparameters across various tasks, I wonder how the performance of UAICRL might be influenced by the selection of hyperparameters for the FTG network. This concern arises from the potential of FTG to either underfit or overfit, which could lead to generating out-of-distribution trajectories and potentially causing a decline in overall performance. Could you explain the process of tuning the hyperparameters for the FTG network?

---

> ### Author Response · Authors · 2023-11-17
> **Author response to Reviewer 4NEv - Part 1**
>
> Dear Reviewer, we sincerely value your time and effort in evaluating our work. We have prepared comprehensive responses and clarifications to address each point you raised. We hope these responses can resolve your concerns.
>
> 1. '*The ablation of the mutual information term in Eq. (7) results in a configuration where only the risk-sensitive constraint from Eq. (4) is utilized. This particular setup is not discussed in the paper.*'
>
> **Response.**  Thank you for your suggestion. Indeed, our ablation study includes a comparison with the UAICRL-NDA baseline, which examines the performance of the model without the consideration of epistemic uncertainty (refer to Table 1 in our paper). Since we estimate epistemic uncertainty using the mutual information term, UAICRL-NDA is structured to study the effect of removing this term. In the paper, we primarily highlight that UAICRL-NDA omits the data augmentation process. This is because, within our experiment, data augmentation has a more significant impact on model performance compared to other components in the mutual information estimation process (entropy and dropout layers as per Equation (8)). Despite this, during the implementation, we remove the entire epistemic uncertainty estimator.
>
> 2. '*I'm concerned about expanding the dataset by generating trajectories based on a learned flow function. Is it still possible for the flow function to generate out-of-distribution data.*'
>
> **Response.**
> Thanks for your comments. Our FTG can generate a rich number of trajectories that are not included in the original dataset (out-of-distribution) due to its ability of:
> - **Generating Diverse Samples.**
> Intuitively, FTG is based on Generative Flow Networks (GFlowNets), which demonstrated excellent performance in generating a diverse array of samples in the realm of scientific discovery ([1]). Rather than consistently producing the highest-scoring trajectories, which correspond to the largest flows, the generation process employed by GFlowNets is characterized by sampling from a distribution represented by learned flow functions. Many of the samples are **new**, and thus they can facilitate scientific discovery.
> - **Generating OoD Trajectories.**
> Conceptually, at each time step, the state-action pairs that are sampled reflect the patterns **observed in the provided dataset** (for example, molecules). However, the true strength of this approach is realized when **these pairs are concatenated**. This process enables the representation of **Out-of-Distribution (OoD) trajectories** (for example, proteins), extending the utility of the GFlowNets beyond the confines of the training data.
> - **Generating OoD Data in Feature Space.**
> Many previous works ([2][3]) have demonstrated the effectiveness of generative models, such as GFlowNets, GAN, and diffusion models, in dealing with out-of-distribution data. Additionally, it's crucial to note that the superior performance of generative models (like GPT) can be attributed to leveraging high-level latent representations of raw inputs. Similarly, we believe that our FTG exhibits comparable expressive power, generating trajectories that resemble the training data in the latent space, even if they are OoD in the state-action feature space.
>
> In conclusion, with efficient representing capabilities and objective-driven sampling, FTG is able to generate OoD data.
>
>
> **References**
>
> [1] Bengio, Yoshua, et al. "Gflownet foundations." Journal of Machine Learning Research (2023).
>
> [2] Jain, Moksh, et al. "GFlowNets for AI-driven scientific discovery." Digital Discovery (2023).
>
> [3] Marek, Petr, et al. "OodGAN: Generative Adversarial Network for Out-of-Domain Data Generation." NAACL-HLT (2021).

---

> ### Author Response · Authors · 2023-11-17
> **Author response to Reviewer 4NEv - Part 2**
>
> 3. '*I'm wondering if the flow functions can be substituted with other conditional generative models, or if the flow matching objective is tightly coupled with UAICRL. For example, can one use a conditional diffusion model to replace the Flow-based Trajectory Generation (FTG) algorithm?*'
>
> **Response.** We have chosen to utilize GFlowNets due to its primary design focus on **generating sequences or trajectories**. In detail, the generative process of the Flow Trajectory Generator (FTG) is **Markovian**, meaning it generates the state at time $s_t$ based on the preceding state $s_{t-1}$.
>
> In contrast, diffusion models are primarily developed for **image generation**. Recent works have expanded these models to generate trajectories [4]. However, the process of diffusion or denoising is **inherently non-Markovian** (unless specific restrictions are imposed), resulting in the generation of an entire trajectory in one go. This model resembles a Bandit solver.
>
> In the context of the Inverse Constrained Reinforcement Learning (ICRL) problem, the sampling of constraint-violating behaviors at any given time step should depend on the previous states and actions. Therefore, while it is possible to replace FTG with a diffusion model, we maintain that the flow-based generator is more **closely aligned with the requirements of our work.**
>
> We have also considered the Decision Transformer (DT) [5]. However, DT is primarily proposed to generate actions instead of states, which makes it less suitable for our task.
>
>
> 4. '*While Table B.1 suggests that FTG can maintain consistent hyperparameters across various tasks, I wonder how the performance of UAICRL might be influenced by the selection of hyperparameters for the FTG network. This concern arises from the potential of FTG to either underfit or overfit, which could lead to generating out-of-distribution trajectories and potentially causing a decline in overall performance. Could you explain the process of tuning the hyperparameters for the FTG network?'*
>
> **Response.** Thanks for raising this concern. To address your concern, we have revised the paper to include the detailed fine-tuning process of FTG in Appendix B.4. Here, we provide a brief overview of it:
>
> The key hyperparameters in the FTG are **1) sample flows $M$**: since we cannot directly calculate the continuous inflows and outflows to perform flow matching, we discretize the flows and match the discretized flows instead by sampling $M$ actions independently and uniformly from the continuous action space. It is crucial for network training with respect to underfitness or overfitness. **2) sample action size $K$**: for generating in continuous spaces, we need to sample $K$ actions from the action space and calculate their corresponding flows, and actions with larger flows will be executed with higher probabilities. Relatively large $K$ will promote exploration in the generation process, which helps to generate diverse trajectories. **3) network architecture**. **4) learning rate**.
>
> Following the hyperparameters in CFlowNets [6], which also study the MuJoCo environment using generative networks, we keep **the sample action size $K$, network architecture and learning rate**, and only adjust the sample flows $M$. Ideally, large $M$ may make it harder for the network to converge, while small $M$ may result in overfitting. In this work we set $M=50$ (instead of $M=100$ in [6]). The reasons are as follows: (1) Too large $M$ will not only make the network hard to converge but also result in a significant increase in GPU memory. To avoid heavy computation and underfitness, we should avoid setting $M$ too large. (2) Too small $M$ will make the network overfit and perform poorly. So we can not set $M$ too small. In addition, we utilized the annealing learning rate to avoid overfitting. (3) We have also **conducted experiments** to help us choose the hyperparameters, including analyzing the quality of the generated trajectories by FTG (check Table B.2) and evaluating the whole performance of UAICRL (check Figure B.1) with different numbers of $M$.
>
> Finally, to trade off the performance and computational load, we set $M=50$ and maintain its consistency without fine-tuning the hyperparameters excessively. As shown in experiments, it could work well across various tasks, demonstrating the FTG's adaptability and robustness. We acknowledge that there is always room for improvement and believe that fine-tuning will be beneficial for specific tasks.
>
> **References.**
>
> [4] Zhu, Zhengbang, et al. "Diffusion Models for Reinforcement Learning: A Survey." arXiv preprint arXiv:2311.01223 (2023).
>
> [5] Chen, Lili, et al. "Decision transformer: Reinforcement learning via sequence modeling." NeurIPS (2021).
>
> [6] Li, Yinchuan, et al. "CFlowNets: Continuous Control with Generative Flow Networks." ICLR (2023).

---

> ### Author Response · Authors · 2023-11-22
> **Invitation for further review feedback**
>
> Dear Reviewer,
>
> We sincerely appreciate the time and effort you have dedicated to reviewing our manuscript. Your insightful and constructive feedback has been invaluable to us. We have addressed your questions in the response and incorporated them in the revised manuscript.
>
> Should there be any further questions or clarifications needed, please do not hesitate to reach out. We are committed to responding promptly and comprehensively within the allowed period.
>
> Warmest regards,
>
>  Authors of Paper 5021

---

> > ### Comment · Reviewer_4NEv · 2023-11-23
> >
> > Dear Authors,
> >
> > Thank you for the clarifications. I will be maintaining my score.

---

> > > ### Author Response · Authors · 2023-11-23
> > >
> > > Thank you for your reply. We are glad that your concerns are addressed.  We believe that our paper is strong enough and can make significant  contribution to ICRL.

---

### Author Response · Authors · 2023-11-17
**Summary of updates**

Dear Reviewers, Area Chairs, and Program Chairs,

We deeply appreciate the valuable feedback and guidance provided, which have been instrumental in enhancing our work. In response, we've incorporated detailed clarifications, expanded explanations, additional experimental results, and more illegible figures into our revised paper (modifications are marked in blue). We summarize our major updates below:

- **Expanded Discussion and Explanation of Experiments**: As suggested by Reviewers nMsK and 8mMN, we have provided a more thorough discussion and explanation of significant phenomena and observations arising from our experiment results. These findings yield critical empirical insights into the advantages of UAICRL by comparison with various baselines. The relevant additions can be found in Section 6.2 (highlighted in blue) and Appendix E.

- **Presentation and Clarity**: In accordance with the valuable suggestions from Reviewer zJng, we have revised the paper to enhance its presentation. This includes summarizing the main experimental questions, revising tables and figures, and improving the conclusion section, among others. We have ensured these modifications substantially enhance the overall clarity of the paper.

- **Fine-tuning of Hyperparameters**: As suggested by Reviewer 4NEv, we have defined the crucial hyperparameters of the FTG, along with their potential impacts on FTG's overall performance. We have detailed the process of tuning these hyperparameters for the FTG network in this work (refer to Appendix B.4).

- **Illustration of Trajectories Generated by FTG**: Upon the suggestion of Reviewer 8mMN, we have visualized the trajectories generated by FTG in the Gridworld scenarios. This visual display offers a clear understanding of FTG's capability to generate diverse trajectories and its role in mitigating the impact of epistemic uncertainty. The corresponding results can be observed in Figure 7 and Figure D.4.

- **Generalizability of UAICRL to Multi-Cost Settings**: In response to the suggestion from Reviewer nMsK, we have extended the HighD environment to incorporate both speed and distance constraints. This adaptation allows us to investigate the model's performance under scenarios with multiple constraints. The relevant analysis can be found in Appendix D.4.

We hope the revising can address the concerns and hope the reviewers can provide us with some valuable feedback, and we are more than ready to engage in the discussion.

---

### Meta-Review · Area_Chair_Q4DM · 2023-12-06

**Metareview:**

The submitted paper considers the problem of inverse constrained reinforcement learning (ICRL), focusing on the development of a method therefore which accounts for aleatoric uncertainty. Concretely, the authors develop the Uncertainty-aware Inverse Constrained Reinforcement Learning (UAICRL) algorithm which accounts for aleatoric uncertainties through distributional Bellman updates of the cumulative cost model. To also improve performance in the face of epistemic uncertainty, a flow-based generative data augmentation approach is proposed. The utility of UAICRL is demonstrated in various environments with stochastic dynamics, in which UAICRL outperforms sensible baselines.

Strength of the paper: The paper addresses a relevant problem that has not been sufficiently studied in previous research, proposes a sensible solution, and demonstrates in extensive experiments the usefulness of the proposed approach and the importance of properly accounting for aleatoric uncertainty in environments with stochastic dynamics.

Weaknesses of the paper: The main weaknesses identified by the reviewers concerned the clarity of the presentation, including questions about the Flow-based Trajectory Generation (FTG), the theoretical motivation for the proposed approach, and the structure and discussion of the experimental results. The authors addressed most of these weaknesses in their rebuttal and corresponding updates to their paper (clarification of FTG capabilities; improved structure of experiments). Other reviewers' concerns regarded the significance and partly incremental nature of the proposed approach.

Overall, in line with the majority of the reviewers' recommendations (3x weak accept, 1 weak reject) I think the merits outweigh the shortcomings of the paper and the paper does a good job in improving the performance of ICRL in environments with stochastic dynamics which can be relevant in several applications. Hence I am recommending acceptance of the paper.

**Justification For Why Not Higher Score:**

An ok paper that deserves to be accepted but also not a game changer.

**Justification For Why Not Lower Score:**

A decent paper that proposes a sensible approach for inverse constrained reinforcement learning better accounting for aleatoric uncertainty than existing methods. So it's worth being published.

---

### Decision · Program_Chairs · 2024-01-16

Accept (poster)